# National Weather Service Alaska Sea Ice Program: Gridded ice concentration maps for the Alaskan Arctic

Astrid Pacini[1], Michael Steele[1], Mary-Beth Schreck[2]

[1]University of Washington Applied Physics Laboratory, Seattle, WA, 98105, USA

5   [2]National Weather Service Alaska Sea Ice Program, Anchorage, AK, 99513, USA

*Correspondence to*: Astrid Pacini (apacini@uw.edu)

**Abstract.** There are many challenges associated with obtaining high-fidelity sea ice concentration (SIC) information, and products that rely solely on passive microwave measurements often struggle to represent conditions at low concentration, especially within the Marginal Ice Zone and during periods of active melt. Here, we present a newly-gridded SIC product for 10  the Alaskan Arctic, generated with data from the National Weather Service Alaska Sea Ice Program (hereafter referred to as ASIP), that synthesizes a variety of satellite SIC and in situ observations from 2007-present. These SIC fields have been primarily used for operational purposes and have not yet been gridded or independently validated. In this study, we first grid the ASIP product into 0.05° resolution in both latitude and longitude (hereafter referred to as gridded ASIP, or grASIP). We then perform extensive intercomparison with an international database of ship-based in situ SIC observations, supplemented 15  with observations from Saildrones. Additionally, an intercomparison between three ice products is performed: (i) grASIP, (ii) a high-resolution passive microwave product (AMSR2), and (iii) a product available from the National Snow and Ice Data Center (MASIE) that originates from the US National Ice Center (USNIC) operational IMS product. This intercomparison demonstrates that all products perform similarly when compared to in situ observations generally, but grASIP outperforms the other products during periods of active melt and in low SIC regions. Furthermore, we show that the similarity in performance 20  among products is partly due to the deficiencies in the in situ observations' geographical distribution, as most in situ observations are far from the ice edge in locations where all products agree. We find that the grASIP ice edge is generally farther south than both the AMSR2 and MASIE ice edges, by an average of approximately 50 km in the winter and 175 km in summer for grASIP vs. AMSR2, and 10 km in the winter and 40 km in the summer for grASIP vs. MASIE.

Key Points:

25     1.  ASIP is an operational, mostly remote sensing-based sea ice dataset for the Alaskan Arctic that has not previously been gridded nor independently validated. Here we describe how the data are read, reformatted, gridded, and validated

against a relatively under-utilized in situ sea ice concentration dataset. These in situ data are also used for validation with two other satellite-based products: AMSR2 and MASIE.

2. All three products considered (grASIP, AMSR2, MASIE) perform similarly when compared against in situ observations when the full SIC range 0-100% is considered.

3. For the Marginal Ice Zone (MIZ; SIC>=20% & <=80%), grASIP performs better than AMSR2 at predicting the presence of ice; MASIE has by definition no information at SIC < 40%. In the MIZ, grASIP tends to over-predict SIC, while AMSR2 under-predicts SIC by a larger amount.

4. We find that the grASIP ice edge is on average farther south than both the AMSR2 and MASIE ice edges, with no systematic differences as a function of longitude in the Alaskan Arctic.

## 1 Introduction

The significant change in Arctic sea ice cover over the last century is a clear indicator of the effects of anthropogenic greenhouse gas emissions on our high-latitude oceans (e.g. Fox-Kemper et al., 2021). Strong reductions in sea ice extent and thickness, changes in ice age, and changes in ice drift and deformation characterize the modern record (e.g., Haine and Martin, 2017; Perovich et al., 2020). Serreze and Stroeve (2015) demonstrated that September sea ice extent is decreasing at a rate of 13.3% per decade over the satellite record. Synchronously, the mean ice thickness has decreased from 3.20 m (1958-1976) to 1.43 m (2003-2007) (Kwok and Rothrock, 2009; Lindsay and Schweiger, 2015; Haine and Martin, 2017). Beyond the physical and ecosystem impacts of these trends, changes in sea ice are altering the operational and research environments for ships transiting the Arctic Ocean and its marginal seas; increased shipping is occurring along both the Northwest Passage and the Northern Sea Route (e.g. Arctic Council, 2009; Boylan, 2021), which in turn necessitates increased infrastructure in the region for safety and governance. At the very basic level, these ships need reliable ice maps to inform routing decisions. However, despite the clear trends in Arctic sea ice and the need to measure these changes, there are many challenges associated with obtaining high-fidelity measurements of sea ice concentration (SIC). This is especially true in low concentration environments and during periods of active melt (e.g. Kern et al., 2020).

Historically, there have been two main approaches to measuring SIC from satellite: passive microwave measurements and the use of other imagery to create synthesized ice maps. The former provides a continuous, near-daily record of ice conditions in polar regions since October 1978. Thus, passive microwave is a powerful tool for diagnosing and analysing the long-term evolution of ice, as a consistent processing algorithm can be applied to a consistent observational record (e.g. Meier et al., 2015). There are numerous algorithms and quality-control methodologies in place to processes these measurements of brightness temperature (e.g. Cavalieri et al., 1984, 1999; Comiso, 1986; Comiso and Nishio, 2008; Lavergne et al., 2019). Detailed comparisons of these algorithms are provided by Ivanova et al., 2014, 2015; furthermore, Ivanova et al. 2015 document potential challenges these algorithms face, including the presence of melt ponds, thin ice, and atmospheric effects. A key challenge to brightness temperature measurements is the inability of these measurements to distinguish between melt

water on the surface of ice, leads in the ice, and open water conditions (e.g. Kern et al., 2020; Meier and Notz, 2010; Gogineni et al., 1992; Grenfell and Lohanick, 1985). This leads to an underestimation of sea ice concentration, which in turn results in an underestimation of sea ice extent during periods of active melt, especially in summer months (e.g. Kern et al., 2020; Ivanova et al., 2015; Rösel et al., 2012b; Markus and Dokken, 2002; Comiso and Kwok, 1996; Steffen and Schweiger, 1991; Cavalieri et al., 1990).

A second method for obtaining ice concentration information from satellite relies on an analyst to synthesize the information in and across different satellite imagery. These maps are drawn by utilizing active sensors such as scatterometers and synthetic aperture radar (SAR), and passive sensors such as visible and microwave imagers. Operational maps are drawn by an analyst using available imagery, but they are not consistent over the historical record due to improvements in satellite technology, the availability of clear imagery (often dependent on cloud and weather conditions), and changes in analyst personnel (e.g. Kern et al., 2020). These human-synthesized ice charts can be higher resolution than their passive microwave counterparts and may better represent summertime conditions owing to their use of multiple satellite products. On the other hand, there are a growing number of products that use the same inputs but produce sea ice concentration maps automatically for research and operations (e.g. Map-Guided Ice Classification developed for the Canadian Ice Survey (Leigh et al., 2014) and AI14Arctic Sea Ice Challenge which merges Sentinel-1 SAR imagery with passive microwave data (Jørgen et al., 2022)). We here define operational ice charts as maps that are intended to serve stakeholders who need SIC information for real-time operation. Often, analyst-generated products provide ice information using polygons instead of a regular grid, and these polygons can be smaller than the footprint of some satellites (e.g. SSMI). That said, these polygons typically provide ice concentration ranges, not specific ice concentration values, and thus provide a quantized ice concentration estimate, rather than a continuous field. As with passive microwave products, there are many processing agencies and techniques used to generate synthesized ice maps. For example, the U.S. National Ice Center produces pan-Arctic daily, weekly, and monthly ice charts as well as domain-specific services upon request. Also, the Danish Meteorological Institute publishes operational ice maps of conditions around Greenland, while the Norwegian Meteorological Institute covers the Atlantic Arctic, including the Nordic Seas, eastern Greenland, Iceland, and Svalbard, the Finish Meteorological Institute reports on the Baltic Sea, the Arctic and Antarctic Research Institute focuses on the Barents, Kara, Laptev, and East Siberian Seas, and finally, the Canadian Ice Service covers the Canadian Arctic.

While passive microwave can provide insight on long term trends, manually-synthesized products can describe in high resolution the daily state of the sea ice pack. For example, in the field, operational stakeholders rely on these data to decide on ship routes and instrumentation deployment locations; in research, these maps provide valuable high-resolution information for summertime conditions and small-scale dynamics (e,g, Chiodi et al., 2021). Synthesized ice maps can also be a valuable tool for driving operational models and weather predictions (e.g. Meier et al., 2015), although most large-scale users rely on low-latency SIC fields from passive microwave (e.g. Chin et al., 2017).

There are a few studies that compare the accuracy of these manually-synthesized ice products with satellite data. For example, Agnew and Howell (2003) compare weekly Canadian Ice Service sea-ice charts (Canadian Ice Service, 2009) from 1979-1996 with passive microwave data derived using the NASA Team algorithm to show that the NASA Team algorithm underpredicts sea ice area in the Canadian Arctic by 20.4 – 33.5% during summer melt periods, and by 7.6 – 43.5% in the fall while ice is growing. Another study by Meier et al. (2015) compares passive microwave measurements against Multisensor Analyzed Sea

Ice Extent (MASIE), a product that originates from the US National Ice Center (USNIC) operational IMS product (Interactive Multisensor Snow and Ice Mapping System, USNIC, 2008), from 2006-2014. They show that MASIE produces a higher-resolution ice edge compared with the passive microwave ice edge, and that sea ice extent from MASIE is generally larger than that derived from passive microwave.

In this study, we present data from an operational SIC product for the Alaskan Arctic generated by the National Weather Service Alaska Sea Ice Program  (hereafter referred to as ASIP). Our motivation comes in part from a case study of conditions in the Beaufort Sea on August 21, 2022 (Fig. 1). The color scale used in all panels of Fig. 1 is World Meteorological Organization standard for ice charts. Details of WMO ice nomenclature can be found in Section 2.1; throughout this study we follow ASIP and international ice chart convention by using the WMO color code and descriptors for characterizing ice

concentration ranges (WMO, 2014; Environment Canada, 2005). For AMSR2 data, we re-binned individual grid cells into the WMO ice concentration ranges. As such, data were rounded to the nearest tens place in concentration e.g., a pixel with ice concentration of 35-64% would fall into the yellow band that represents 4/10 to 6/10 ice. During this time, the NASA Salinity and Stratification at the Sea Ice Edge (SASSIE) field program was operating in the Pacific Arctic (Drushka et al., 2024). As part of the field program, four remotely operated Wave Glider vehicles (Thomson, 2023) were deployed to obtain

measurements of salinity near the sea ice edge. To do this successfully, there was a critical need for accurate daily ice edge information. Figure 1a depicts the ice concentration on August 21, 2022 from a 25 km resolution passive microwave dataset (AMSR2) that relies on the NASA Team 2 algorithm, obtained from the USNIC (Markus et al., 2018). This date was chosen in order to represent conditions in the middle of the Wave Glider deployment. This product shows compact ice north of 72°N, with pockets of open water found at 73°N, 150°W and 74.5°N, 146°W and open water west of 157°W and south of 72°N.

AMSR2 data processed using the ASI algorithm (Fig. 1b; Spreen et al., 2008) at 3.125 km resolution show a similar ice distribution as in Fig 1a, although with higher spatial detail (including larger areas of open water within the pack). We compare these two passive microwave measurements with MASIE (Fig. 1c; US National Ice Center et al., 2010), which uses a binary ice flag to indicate either ice or open water and a 40% ice concentration cut-off (see details in section 2.3.2).  Consistent with both AMSR2 products, MASIE indicates compact ice north of 72°N, although MASIE indicates the presence of ice to the west

of 157°W. MASIE also predicts a region of ice to the east of Prudhoe Bay, extending south from the icepack toward Alaska near 71°N. Finally, ASIP (Fig. 1d) shows ice throughout the domain north of 72°N, consistent with MASIE. However, ASIP

also identifies significantly more low-concentration ice at the southern boundary of the ice pack, including a tongue of ice extending toward the southeast in the direction of Prudhoe Bay. This tongue is not present in the three other products considered.


The positions of the four Wave Gliders for one week prior to, and one week after, these maps are shown in gray in Fig. 1. Images taken during the Wave Glider deployment cruise onboard the RV Ukpik and mission-specific support from the USNIC (which included visible images of ice conditions) demonstrated that ice was clearly present during the deployment, in agreement with ASIP and in disagreement with AMSR2 and MASIE. Furthermore, the presence of ice during deployment
resulted in two Wave Gliders being deployed to the west of the tongue of ice at approximately 150°W and two to the east of this tongue. Despite efforts to join these tracks over the two weeks shown in the figure, the persistence of this ice tongue resulted in two separate survey regions. Therefore, it is clear that in this case, ASIP best represented the presence of low concentration ice near the Alaskan coast. A second motivation for investigating the use of ASIP was its superior performance in evaluating sea ice during a NASA-sponsored saildrone cruise in the northeast Chukchi Sea, also during summer 2022 (not
shown here; see García-Reyes et al., 2023). These two NASA-sponsored field campaigns, in addition to the previously discussed increase in vesssel activity near the Arctic ice pack, motivates this detailed investigation of the performance of ASIP in the Pacific Arctic more broadly.

The paper is structured as follows. First, we describe the data reading, reformatting, and gridding methodology (ultimately
producing a gridded ASIP, or what we refer to as grASIP), then we perform an intercomparison with in situ measurements of sea ice, compare grASIP to a passive microwave product (AMSR2) and a second product (MASIE), and finally investigate the location of the ice edge in all three products.

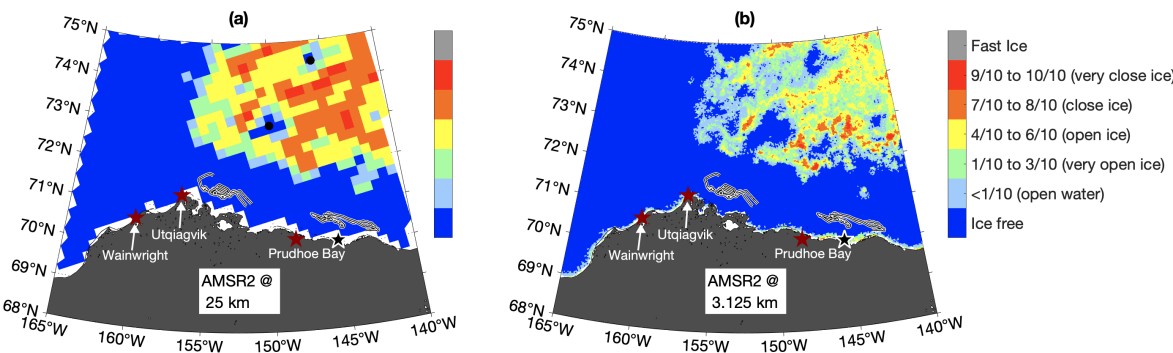

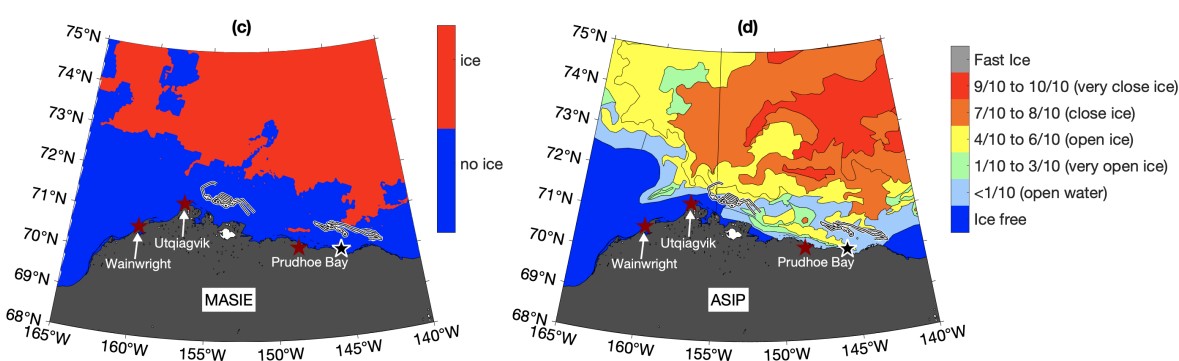

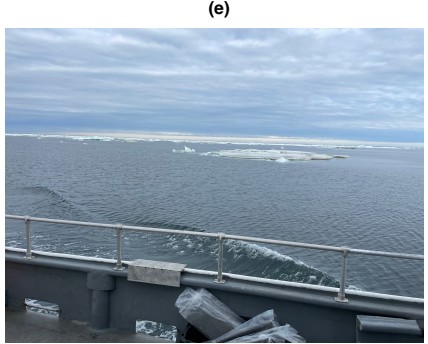

Figure 1: Sea ice conditions according to different ice products for August 21, 2022. (a) AMSR2 at 25 km resolution using the NASA Team 2 algorithm (Markus et al., 2018). (b) AMSR2 at 3.125 km using the ASI algorithm (Spreen et al., 2008). (c) MASIE ice/no ice (US National Ice Center et al., 2010). (d) National Weather Service Alaska Sea Ice Program. The track lines of four Wave Gliders operating in the region are overlaid in white, representing the tracks for one week prior to one week post the ice images (Aug 14 – Aug 28, 2022). (e) Image taken by Jim Thomson during the Wave Glider deployment cruise on August 12, 2022. Location of the image is marked by the black star in (a)-(d).

## 2 Data

### 2.1 National Weather Service Alaska Sea Ice Program

The National Weather Service's Alaska Sea Ice Program provides real-time ice information for conditions in the Pacific Arctic (Heim and Schreck, 2017). This product is primarily operational; ASIP supports ships working in the region, including shipping and transportation vessels, the fishing fleet, the Coast Guard, tourist vessels, and research vessels. ASIP also supports Alaska native communities and subsistence hunts, the oil and gas industry, Alaska Fish & Wildlife, the Department of Homeland Security, and National Weather Service forecasters (Hufford, 2009; Deemer et al., 2017). ASIP issues a variety of products to stakeholders, including text-based and graphical information. Specifically, ASIP issues daily sea ice maps for the full domain (135°W – 175°E, 45°N – 80°N) and regional sectors, a 5-day graphical sea ice forecast, text-based sea ice forecasts for five days into the future, including regional forecasts for the Beaufort, Chukchi, and Bering Seas, and a three-month sea ice outlook for the entire domain. Additionally, ASIP publishes a sea surface temperature analysis for the region.

The data are produced following World Meteorological Organization (WMO) standards for archiving digital ice charts (WMO, 2010). Specifically, all ice maps are presented as GIS Shapefiles following SIGRID-3 vector archive format. ASIP creates daily SIGRID-3 ice charts for Alaskan waters including the Beaufort Sea, Chukchi Sea, Bering Sea, and Cook Inlet. These SIGRID-3 files have two main components: the shapefile containing the ice analysis information (ice polygons and related attributes) and the metadata describing the ice analysis data under the SIGRID-3 format. The Sea Ice Analysis represents ice conditions valid at approximately 0030 UTC and is based on imagery acquired over the preceding 24 hours. Imagery utilized includes synthetic aperture radar (SAR), Polar Orbiter Satellite (POES), visible and infrared imagery such as Visible Infrared Imaging Radiometer Suite (VIIRS), other optical or infrared sensors prioritized by latency and image quality, and local observations and forecast weather conditions. ASIP data include published ice charts three times per week (Monday, Wednesday, and Friday) from 2007 – June 2014 and daily data from July 2014 – present. Prior to March 2007, ASIP published hand-drawn charts which are not included in this analysis. As the USNIC also produces operational ice maps covering this region, one might ask whether these products have been compared. ASIP and USNIC are collaborators, but they have independent data streams and these two ice maps have not been compared in the scientific literature.

WMO standard for digital ice charts stipulates that data be encoded following a set of concentration and specified colors (WMO, 2015, Environment Canada, 2005). Each polygon of ice information contains alphanumeric information on SIC, stage of development and thickness, and ice form and floe size. SIC information is expressed in tenths: areas with no concentration information are ice-free, a concentration value of less than one-tenth is called open water/bergy water, 1/10-3/10 indicates "very open ice", 4/10-6/10 represents "open ice", 7/10-8/10 represents "close ice", 9/10-10/10 indicates "very close ice", and a fast ice category is also encoded. For stage of development information, alpha-numeric codes correspond to different ice descriptors (e.g. Code 1 indicates new ice <10 cm, Code 6 indicates first-year ice >= 30 cm, etc.). A similar system is employed

for ice form and floe size (e.g. Code 0 indicates pancake ice, Code 6 indicates vast floe (2-10 km), etc.). In this manuscript we focus only on SIC, but we note that the reading, reformatting, gridding, and validating presented here could also be done for stage/thickness and form/floe size. WMO also defines a standard color bar to indicate these concentration ranges (e.g. Fig. 1, Fig. 2a,d). It is important to note that WMO, and in turn ASIP, call <1/10 concentration "open water/bergy bits," which is not the nomenclature that will be adopted in the remainder of this study. From this point on, open water refers to conditions where no ice is present (WMO refers to this as ice free).

A polygon is generated by an individual analyst. Available imagery for the preceding 24 hours is visualized, and the analyst uses these data to manually select the contour representing the bounds of a given polygon. Generally, polygons are drawn around ice that appears homogenous, or ice floes that look relatively evenly distributed. On any given day, up to approximately 300 polygons are specified, with smaller, higher-resolution polygons generally near the ice edge and larger polygons within the ice pack. The daily number of polygons specified varies seasonally, with more polygons drawn in fall, winter, and spring and fewer polygons drawn in late summer, when little ice is present across the domain. Additionally, the number of polygons has steadily increased over time, coincident with an improvement in the resolution of the imagery used to generate ASIP maps. Fig. 2a,d illustrate winter and summer polygon examples.

The first step in our procedure is to read the data.. The concentration and alphanumeric codes are converted to numerical values (i.e. the character string associated with a polygon (e.g. 68) is converted to 6/10 – 8/10). Subsequently, the data are gridded. Here, we choose a 0.05° grid (in both latitude and longitude) with no interpolation. For a given ASIP polygon, each grid point that lies within its boundary is assigned that polygon's ice concentration value. The choice of 0.05° is made in order to resolve the smallest polygons, without rendering the data set too large, as there is no native grid or resolution to the polygons themselves. Polygons embedded fully within other polygons present a challenge to this gridding algorithm, as a choice must be made as to which polygon takes precedence. In this study it is stipulated that a smaller polygon will always supersede a larger polygon; this means that if a smaller polygon is embedded in a larger polygon, then for the spatial extent of the smaller polygon, the value of that smaller polygon is utilized, and for the remainder of the large polygon, the value of the larger polygon is utilized. To obtain a single value of SIC from the range of concentrations identified by the analyst (Table 1), we take the mean value of the range (see Fig. 2b,e), and present the range divided in half as error bars (e.g. a polygon coded as 1/10 – 3/10 ice would have 20% SIC, with 10% error bars) (see Fig 2c,f). Similar methodology has been utilized for other datasets converted from SIGRID format into gridded numerical fields (e.g. USNIC Arctic and Antarctic Sea Ice Concentration and Climatologies in Gridded Formation, USNIC, 2020). From this point forward, grASIP will refer to our gridded SIC data set and thus differs from the source data published by the National Weather Service (https://www.weather.gov/afc/ice).

It is important to note that there are only 11 concentration ranges used by ASIP polygons (Table 1), which means that our gridded product also has only these 11 discrete values. Alternatively, one could imagine producing a spatially smoothed version

of this dataset using a fixed or perhaps variable length scale. This is beyond the scope of this paper, as our goal here is simply to ingest and reformat, grid, and validate the raw grASIP data, and to compare these data with other ice concentration products.

220 That said, if these data were to be ingested into climate models or weather forecasts, then continuous fields could be of interest. Therefore, efforts to produce a smoothed product are ongoing.

| Ice concentration range | SIC (%) |
|---|---|
| | 0 |
| <1/10 | 5 |
| 1/10 – 3/10 | 20 |
| 2/10 – 4/10 | 30 |
| 3/10 – 5/10 | 40 |
| 4/10 – 6/10 | 50 |
| 5/10 – 7/10 | 60 |
| 6/10 – 8/10 | 70 |
| 7/10 – 9/10 | 80 |
| 8/10 – 10/10 | 90 |
| 9+ (meaning >9/10 | 95 |
| 10 (shorefast ice only) | 100 |

**Table 1: WMO standard ice concentration ranges, with corresponding average SIC.**

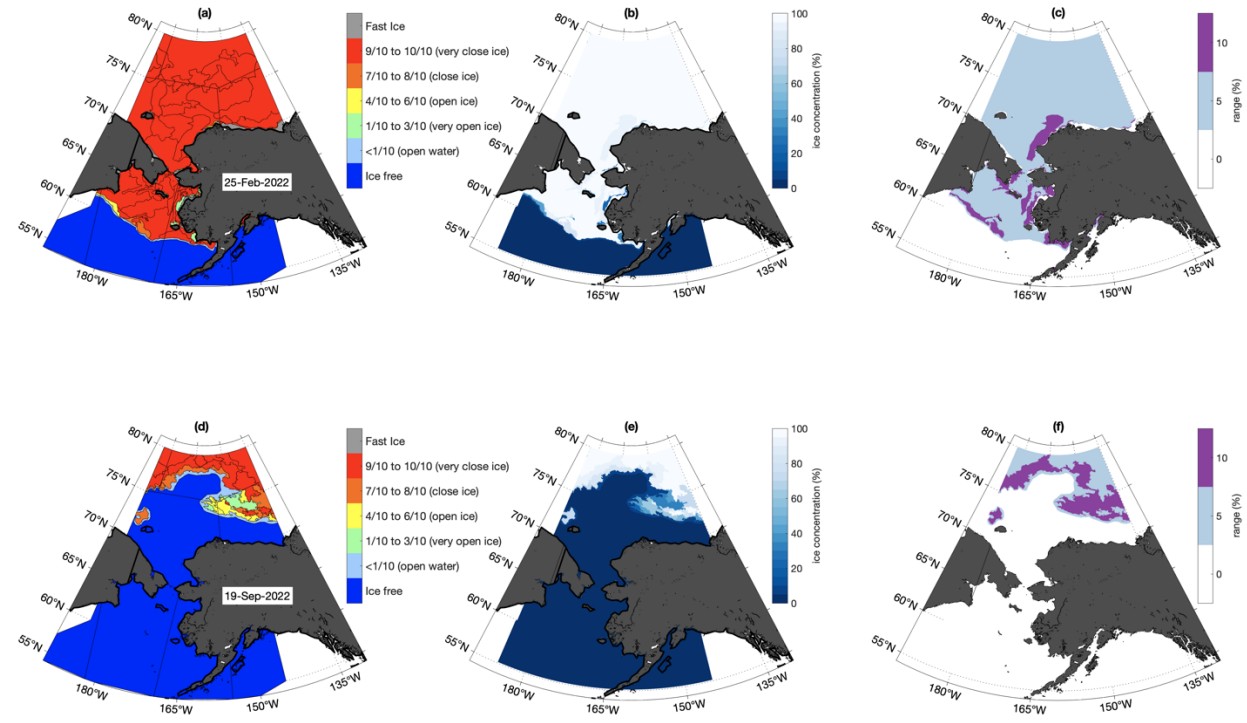

**Figure 2: Example NWS ASIP polygons and resulting gridded product (grASIP) for example winter (top row; 25 Feb, 2022) and summer (bottom row; 19 Sept, 2022) conditions. (a,d) Raw ice polygons derived from shapefiles that indicate WMO standard primary ice type concentration. (b,e) Gridded SIC for corresponding day, see text for details on ice concentration calculation. (c,f) SIC error bars, calculated as described in the text.**

## 2.2 In Situ Observations

We use a total of 5991 in situ observations from the Pacific Arctic for the years 2007-2022, which include ship-based observations and Saildrone measurements (Table 2 and Fig. 3a). Most of these observations were collected during summer months (Fig. 3b) with a relatively even distribution among years (with the exception of 2012 and 2021, which had low sampling

rates) (Fig. 3c). These in situ observations span a range of SIC, with the majority sampling either open water or compact ice (Fig. 3d). Note that the Saildrone observations are not included in Fig. 3d, because these only provide binary ice/no ice information (Table 2) and therefore cannot be binned into concentration ranges.

| Observational Platform | Dates | Observation type | Data source |
|---|---|---|---|
| CCGS Louis S. St-Laurent | 26 July – 31 Aug, 2007 | % | Ice Watch |
| CCGS Louis S. St-Laurent | 17 July – 20 Aug, 2008 | % | Ice Watch |
| CCGS Louis S. St-Laurent | 17 Sept – 15 Oct, 2009 | % | Ice Watch |

| | | | |
|---|---|---|---|
| CCGS Louis S. St-Laurent | 15 Sept – 15 Oct, 2010 | % | Ice Watch |
| CCGS Louis S. St-Laurent | 21 July – 18 Aug, 2011 | % | Ice Watch |
| USCGC Healy | 15 Aug – 28 Sept, 2011 | % | Ice Watch |
| CCGS Louis S St-Laurent | 1 Aug – 8 Sept, 2012 | % | Ice Watch |
| CCGS Louis S St-Laurent | 1 Aug – 2 Sept, 2013 | % | Ice Watch |
| USCGC Healy | 12 May – 23 June, 2014 | % | Ice Watch |
| CCGS Louis S. St-Laurent | 21 Sept – 17 Oct, 2014 | % | Ice Watch |
| USCGC Healy | 9 Aug – 12 Oct, 2015 | % | Ice Watch |
| CCGS Louis S. St-Laurent | 18 Sept – 18 Oct, 2015 | % | Ice Watch |
| RV Sikuliaq | 1 Oct – 10 Nov, 2015 | % | Ice Watch |
| CCGS Sir Wilfrid Laurier | 1 July – 22 July, 2016 | % | ArcticData |
| CCGS Louis S. St-Laurent | 22 Sept – 16 Oct, 2016 | % | Ice Watch |
| USCGC Healy | 26 Aug – 15 Sept, 2017 | % | ArcticData |
| RV Araon | 3 Aug – 26 Aug, 2018 | % | Ice Watch |
| CCGS Louis S. St-Laurent | 5 Sept – 2 Oct, 2018 | % | Ice Watch |
| USCGC Healy | 14 Sept – 19 Oct, 2018 | % | Ice Watch |
| Saildrones (x3) | 15 May – 11 Octo, 2019 | binary | Chiodi et al., 2021 |
| CCGS Louis S. St-Laurent | 12 Sept – 15 Sept, 2019 | % | Ice Watch |
| RV Sikuliaq | 7 Nov – 27 Dec, 2019 | % | Ice Watch |
| KV Svalbard | 15 Oct – 25 Nov, 2020 | % | Ice Watch |
| Saildrones (x2) | 18 June – 17 July, 2022 | binary | García-Reyes et al., 2023 |

**Table 2: In situ observations with platform, survey duration, observation type (either binary ice/no ice or SIC %), and data source.**

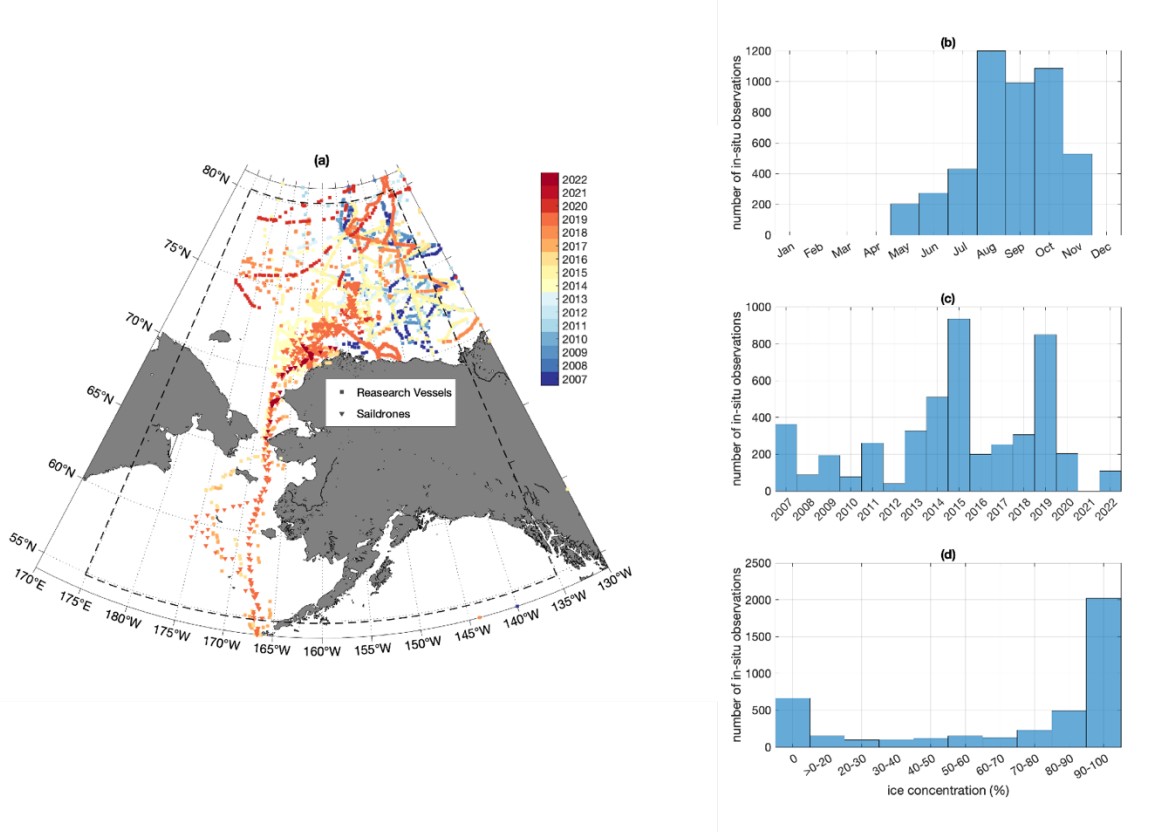

**Figure 3: In situ observations. (a) Map of in situ observations, shaded by year of observation. (b) Histogram of seasonal distribution of in situ observations. (c) Histogram of interannual distribution of in situ observations. (d) Histogram of SIC distribution of in situ observations.**

### 2.2.1 Ship-based observations

The ship-based observations are predominantly obtained from the international Ice Watch program. This program follows Antarctic Sea Ice Processes & Climate protocol (ASPeCt) (e.g. Worby and Allison, 1999; Worby and Dirita, 1999) extended to the Arctic through the software for Arctic Shipborne Sea Ice Standardization Tool (ASSIST) (Hutchings, 2020). This

program facilitates standardized collection of ship-based human observations of ice and meteorological information. Generally, Ice Watch protocol requires ice observations to be made within 1 nautical mile of the ship with 360° visibility during a 10-minute sampling window. The Ice Watch program, and the ASSIST software specifically, provide instructions for determining parameters like total ice concentration, open water amount, snow, topography, melt, and other information including ice type, thickness, and floe size. This terminology comes from ASPeCt protocol, which in turn is derived from

WMO codes, and is used to maintain consistency between the hemispheres. Previous studies have evaluated in situ observations of ice to validate satellite ice edges, especially in the Antarctic (e.g. Worby and Comiso, 2004; Bietsch et al.,

2015), and recent studies by Kern et al. have evaluated ice products using in situ observations globally (Kern et al., 2019) and within the Arctic (Kern et al., 2020). In this study, we utilize all Ice Watch data within the Pacific Arctic, as defined in Fig. 3, from 2007-2022 (see Table 1 for details). Worby and Comiso (2004) assess an accuracy of 5-10% for Ice Watch SIC, computed by comparing the range of estimated SIC recorded by simultaneous observations made by different observers. Unfortunately, there were not enough instances of simultaneous SIC estimates available to perform the same calculation in the Pacific Arctic. Specifically, there were no Ice Watch observations from different ships occurring within 24 hours of each other, less than 0.25° apart in latitude and longitude.

While most ship observations utilized in this study are obtained through the Ice Watch program, a few cruises conducted ice watches without reporting the data through ASSIST. As specified in Table 2, we utilize 20 Ice Watch cruises that entered the Pacific Arctic during the period of interest (2007-2022) and three cruises that reported ice data to the Arctic Data Center. Two of these cruises (CCGS Sir Wilfred Laurier 2016 and USCGC Healy 2017) provided ice concentration information through the Marine Mammal Watch that is a part of the National Science Foundation's Distributed Biological Observatory program (Moore, 2019a, Moore 2019b, Moore and Grebmeier, 2018). The final non-Ice Watch cruise followed Ice Watch observation protocol for the Study of Under Ice Blooms in the Chukchi Ecosystem program (Polashenski, 2016). For the purposes of this analysis, all in situ SIC data are averaged onto a daily grid centered at midnight UTC, since most ice maps are valid at or near this time stamp. This avoids biasing the analysis due to ships reporting at different temporal frequencies, although it also reduces the database size and effective spatial resolution for ships transiting through the MIZ . Following Kern et al. 2019, if a 24-hour window on a given ship has fewer than three observations, it is discarded and those data are not utilized in the intercomparison. This daily averaging reduces the number of in situ observations to 896.

### 2.2.2 Saildrones

Uncrewed Surface vehicles (USVs) facilitate the collection of high-resolution surface ocean information and meteorological parameters near the air-sea interface. In this study, we utilize two USV campaigns to the Pacific Arctic performed by Saildrones (Cokelet et al., 2015; Meinig et al., 2015; Mordy et al., 2017; Gentemann et al., 2019; García-Reyes, 2023). Saildrones are wind-driven platforms with solar-powered instrumentation. In this study, we rely on Saildrones outfitted with cameras to detect the presence or absence of ice. The Saildrones were outfitted with wing-mounted cameras ~5 m above the surface that captured images in three directions every 5 – 60 min: upwards (sky), downwards (hull and immediate surroundings), and horizontally. The footprint of saildrone camera imagery (~0.3 nm radius, based on Fig 9 in Chiodi et al., 2019) is generally smaller than that provided by human observations of sea ice from ships (~ 1 nm radius).  Readers are referred to Chiodi et al., 2021 for details on the conversion of image files to a timeseries of ice/no ice from the Saildrone tracks. We utilize the data presented in Chiodi et al., 2021 for three Saildrones operating in the Pacific Arctic in 2019, and we perform the same analysis on the imagery from two Saildrones in 2022. The Saildrone data (2019 and 2022) are averaged on a daily grid with binary ice/no ice information only, with the stipulation that ice must be present for at least 25% of the day to constitute the Saildrone being in ice.

Furthermore, the conditions encountered by the Saildrones are unlikely to be SIC > 40% because the vehicles themselves are
not meant to operate in the ice.

## 2.3 Other ice concentration data sets

It is of interest to compare grASIP data with other satellite-based measurements of SIC in the region. For this analysis, we utilize a high-resolution passive microwave product (AMSR2) and a product from the National Snow and Ice Data Center (MASIE), both of which are used often in the scientific literature. AMSR2 at 3.125 km resolution was chosen in order to
compare grASIP with a high-resolution, solely passive microwave-based product. MASIE was chosen, instead of the USNIC IMS operational product, for example, because it offers a unique daily high-resolution map of ice extent, is provided in an easy-to-use gridded format, and represents a product commonly used in the scientific literature that is generated following similar methodology to the grASIP dataset.

### 2.3.1 Passive microwave (AMSR2)

Advanced Microwave Scanning Radiometer 2 (AMSR2) is a multi-frequency passive microwave sensor onboard the JAXA Global Change Observation Mission-Water (GCOM-W1) satellite. This sensor is a follow-up to AMSR-E which was onboard the NASA satellite Aqua from 2002-2011. AMSR2 measures brightness temperatures at a number of different frequency channels, using vertical (V) and horizontal (H) polarization; the 89 GHz channel provides the smallest footprint and highest resolution among these channels. Because AMSR-E and AMSR2 channels differ, AMSR2 brightness temperatures are then
converted to AMSR-E brightness temperatures using a series of empirically-derived conversion coefficients to maintain consistency between datasets.

The Arctic radiation and turbulence interaction study (ARTIST) sea ice (ASI) algorithm was developed for Special Sensor Microwave Imager (SSM/I) data (Kaleschke et al., 2001) and adopted to AMSR-E data (Spreen et al., 2008) to compute SIC
from brightness temperatures. ASI relies on the difference between the V and H polarizations at 89 GHz to convert brightness temperature into SIC using tie-points and a series of weather corrections that rely on the 18, 23, and 37 GHz frequency channels. Subsequently, SIC is computed using the Bootstrap algorithm (Comiso, 1986). Further algorithm details can be found in Melsheimer, 2024. Why choose this algorithm over other AMSR2 products? Many studies have compared the performance of ASI-derived SIC to other SIC algorithms and data sources (e.g. Wiebe et al., 2009; Heygster et al., 2009;
Ivanova et al., 2015). For example, Beitsch et al. (2014) compared ASI at 3.125 km with ASI at 6.25 km and 12.5 km, as well as SIC from Bootstrap, to SIC observed by MODIS images. They found that ASI at 3.125 km was better able to capture the size and structure of leads in the sea ice when compared to the other listed dataset; thus, the 3.125 km resolution product is an ideal dataset to compare with the other high-resolution SIC products used in this study. Note that AMSR2 at 25 km resolution using the NASA Team 2 algorithm (Markus et al., 2018) shown in Fig. 1a is not considered in the remainder of this analysis.

The swath SIC data are then gridded onto daily Arctic and Antarctic polar stereographic grids that align with the National Snow and Ice Data Center's grids (at 6.25 km and 3.125 km). In this study, we utilize daily 3.125 km data from 2012-2022 obtained from the University of Bremen data archive (Melsheimer and Spreen, 2019; https://data.seaice.uni-bremen.de/amsr2/asi_daygrid_swath/n3125/). Based on error propagation due to sources of uncertainty in radiometric

measurements, tie points, and atmospheric contributions, the errors are estimated to vary across concentration ranges; the data exhibit a 5.7% error at 100% SIC which increases to 25% error at 0% SIC (Spreen et al., 2008). To match the resolution of grASIP, we regridded the AMSR2 data onto a 0.05° latitude/longitude grid for the analysis in section 4.3.

## 2.3.2 Multisensor product (MASIE)

Multisensor Analyzed Sea Ice Extent (MASIE) is a product available from the National Snow and Ice Data Center (U.S.
National Ice Center et al., 2010). MASIE originates from the US National Ice Center operational Interactive Multisensor Snow and Ice Mapping System (U.S. National Ice Center, 2008; Helfirch et al, 2007), valid at 00:00 UTC. The data are then converted to gridded ice maps by NSIDC. It is important to note that while USNIC and IMS are operational centers, NSIDC is not. Accordingly, while the source data for MASIE are operational SIC maps, MASIE itself is not an operational product, as defined in Section 1. Similar to ASIP, to generate IMS maps at the source of the MASIE product, human analysts consider available

imagery from synthetic aperture radar, visible/infrared imagery, passive microwave measurements, and scatterometer data to generate these maps. These binary maps of ice/no ice are generated at both 1 km and 4 km resolution and use a cut-off threshold of 40% SIC, meaning that grid cells with greater than 40% SIC are designated as having ice, and grid cells with less than 40% SIC are considered ice-free (US National Ice Center et al., 2010). In this study, we utilize daily 4 km data from 2007-present obtained from the National Snow and Ice Data Center (https://nsidc.org/data/g02186/versions/1). To match the resolution of

grASIP, MASIE data are re-gridded onto a 0.05° latitude/longitude grid for the analysis in section 4.3.

## 3 Methods

### 3.1 Parity analysis

We compare the three SIC products (grASIP, AMSR2, MASIE) to in situ observations to assess errors in the former, assuming the in situ observations are accurate. For each SIC product, the grid cell nearest the in situ observation in latitude, longitude,

and time is queried. If the time gap between the nearest SIC map and the in situ observation exceeds 12 hours, the comparison is not made. The SIC grid cell is then converted to a binary ice/no ice value and compared against the in situ observation, also converted to a binary ice/no ice value. The motivation for, and methodology behind, converting SIC to binary values is explained in 3.2. Unless otherwise stated, for grASIP this meant that for SIC greater than or equal to 20% ice was considered present, and for SIC less that 20%, ice was considered absent. The same was done for AMSR2. MASIE is already binary, so

a conversion to binary was not necessary.

Confusion matrices are a tool often used in the machine learning literature to evaluate model performance (e.g., Sammut and Webb, 2011); here we adapt the method to assess the performance of our three SIC products. Confusion matrices focus on the number of correct and incorrect matchups between two datasets; in this case, the matchups between in situ observations (taken as truth) and gridded SIC observations are tallied. Within these matrices, correct matchups lie along the parity (i.e., 1:1) line; incorrect matchups fall in the upper left and lower right quadrants. We thus prefer to use the term "parity analysis" in the following analysis.

For each matchup (e.g. grASIP vs. AMSR2 vs. MASIE), we first define the intersection of all three data sets with respect to three variables: (i) years covered (e.g., 2012-2022 for the 3-way comparison, since AMSR2 only starts in 2012), (ii) minimum SIC threshold (i.e., > 40%, from the MASIE criterion), and (iii) binary ice vs. no ice (i.e., converting the SIC from grASIP and AMSR2 into the MASIE ice/no ice framework). Unfortunately, these constraints result in a reduction in matchups by 75% for the 3-way comparison (grASIP vs. AMSR2 vs. MASIE) (Table 3). This is because the Saildrone observations (which represent a high percent of the original in situ data set) were discarded for this analysis, as Saildrone data are considered binary for the 20% SIC threshold, but not for the 40% SIC threshold (see 2.2.2 for details on this).

**3.2 Motivation for and conversion to binary ice/no ice**

A binary ice/no ice framework is used throughout most of this study, in order to assess the ability of grASIP, AMSR2, and MASIE to detect the simple presence or absence of ice. The in-situ SIC data are converted from SIC to a binary value by specifying that all grid cells with SIC greater than or equal to 20% (or 40%, depending on the comparison) are considered ice pixels, while grid cells with less than 20% (40%) SIC contain no ice. Similarly, grASIP and AMSR2 data are converted to binary ice/no ice information following the same methodology.

With the exception of section 4.1.2, our comparisons between ice maps and in situ observations are performed in this binary ice/no ice framework, which we feel is best for two main reasons. First, human observations of SIC are subjective. While Worby and Comiso (2004) quote an uncertainty of 5% - 10% on human-made SIC observations, this is likely an underestimate. As Kern et al. (2019) describe, untrained or less experienced observers will be able to assess SIC at low concentrations (<30%) and at high concentrations (> 80%) relatively easily; however, human observers will naturally struggle in the SIC ranges between these extremes, which is a region of particular interest in this study. Even among experienced observers, there will be discrepancies in the identified SIC exceeding 10%. Further complication and error is introduced because of constraints on visibility, speed of the ship, distribution of floes and preferential navigation to avoid ice, or to raft to an ice floe, and more. Our second reason for choosing a binary framework has to do with the ranges of SIC provided by ASIP, which are often order 20%. This means that a grid cell reporting 80% could actually exhibit 70% or 90% SIC and still fall in that particular classification range. Additionally, ASIP polygons and ranges are assigned by human observers with variable input data, and

thus subjectivity is introduced in the designation of such polygons. When coupled, these two reasons can result in an error
range of 40%, almost half of the SIC range.

## 3.3 Defining the ice edge

In this study, we define the ice edge as the location of either the 20% or 40% ice concentration contour. The 15% SIC contour is a common choice for defining the ice edge in passive microwave products (Zwally et al., 1983; Cavalieri et al., 1991; Meier and Stewart, 2019), and generally falls in WMO's ice free and open water categories (see Fig. 1). Here, we choose 20% because
grASIP will never report 15% SIC (Table 1). The ice edge in this study is identified by using the gridded SIC maps to draw a contour corresponding to the chosen threshold. Of course, these gridded products, as described previously, are derived following differing methodologies. For grASIP, the data are first presented as polygons and then gridded to 0.05° resolution, after which point the ice edge contour is identified. For AMSR2, we re-grid the source data onto the 0.05° latitude/longitude domain and again use the gridded fields to draw a contour and obtain the daily ice edge. For MASIE, we re-grid the data onto
the 0.05° grid; since we re-grid MASIE, this interpolation introduces non-binary values along the ice edge. To remain as true as possible to the source data, the 0.5 contour is chosen as representing the ice/no ice boundary, which accurately recovers the original ice edge. It is important to note that for MASIE, we can only represent the 40% ice concentration contour.

## 3.4 Footprint size

It is important to keep in mind that the footprint size and shapes of the products compared in this study vary. For ASIP, the
analyst utilizes satellite imagery and draws a polygon around ice that appears homogenous or ice floes that are generally evenly-distributed. Therefore, each polygon has a different shape and size. As described previously, AMSR2 synthesizes a range of footprint sizes from different frequency channels to obtain a final grid size of 3.125 km. Similar to ASIP, the USNIC analyst that generates the IMS product at the root of MASIE grids all satellite data onto a ~4 km (and 1 km starting in 2014) grid and then relies on all available ice data to identify which grid cells have 40% or more ice. The ASSIST in situ observations
are for a 1 nm radius circle around the ship, and the frequency of the observations vary. In this study, the in situ observations are averaged to become daily values; thus, in some situations (when the ship is moving) this results in an extension of the spatial scale of the data coverage. Saildrone data exhibit the smallest footprint, with high temporal resolution (again, averaged into daily values). As such, while the comparison performed in the remainder of this analysis treats the SIC observations from the individual products without considering footprint size, it is important to remember that the methodologies behind each
dataset result in inherent differences in footprint sizes. The potential impacts of these differences are explored in the Discussion.

# 4 Results

## 4.1 Ice maps compared with in situ observations

### 4.1.1 grASIP

The parity calculation, as described in section 3., for grASIP data from 2007-2022 is presented. We note that Saildrone observations are binary: the image either includes ice or does not include ice. For this calculation, these binary Saildrone camera observations of ice are taken to be SIC > 20% and thus included as a positive encounter with ice. See Section 3.1.4 for further discussion on Saildrone data use (and Fig. 1 of Chiodi et al., 2019 for example images of ice encounters).

As shown in Fig. 4 and Table 3, grASIP correctly identifies the presence of ice with an accuracy of (535+174)/(535+174+10+33)*100 = 94.3% when compared with the in situ observations. grASIP overpredicts ice 1.3% of the time and under-predicts ice 4.4% of the time. Since this dataset contains more ice-free than in-ice matchups (known as a class imbalance in the confusion matrix literature), we also include the weighted average accuracy. This is done by computing the accuracy for the no ice condition, which represents the specificity (or true negative rate), (Q1/(Q1+Q3)) and the accuracy for

ice conditions, which represents the sensitivity (or true positive rate), (Q4/(Q2+Q4)) and then averaging these two values together. In this case, the weighted accuracy is 91.1% (Table 3); again, it is important to note that this accuracy estimate is only in relation to the available in situ observations, which are not domain-wide and do not cover all times and ice conditions.

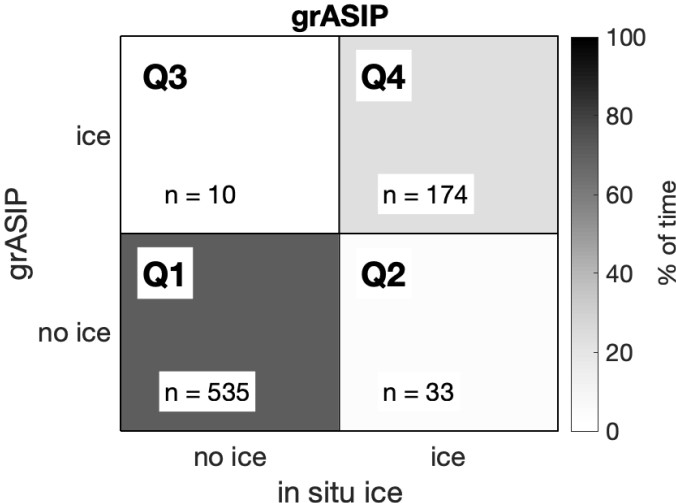

**Figure 4: Full parity plot for ASIP data, 2007-2022.**


| | Datapoints (#) | Accuracy (%) | Weighted Accuracy (%) | Q1 | Q2 | Q3 | Q4 |
|---|---|---|---|---|---|---|---|
| grASIP | 752 | 94.3 | 91.1 | 535 | 33 | 10 | 174 |
| **AMSR2** | 747 | **92.2** | **85.8** | **544** | **57** | **1** | **145** |
| *MASIE* | 301 | *90.7* | *84* | *50* | *8* | *20* | *223* |
| grASIP vs. **AMSR2** | 714 | 95.2, **92.7** | 91.9, **85.0** | 534, **542** | 24, **51** | 9, **1** | 146, **120** |
| grASIP vs. *MASIE* | 228 | 87.3, *89.9* | 83, *85.3* | 45, *46* | 13, *8* | 16, *15* | 154, *159* |
| grASIP vs. **AMSR2** vs. *MASIE* | 190 | 90.5, **86.8**, *90.0* | 86.5, **89.6**, *86.7* | 43, **54**, *44* | 5, **23**, *7* | 13, **2**, *12* | 129, **111**, *127* |
| grASIP vs. **AMSR2** in (0%-40%) | 28 | 42.9, **21.4** | | | 16, **22** | | 12, **6** |
| grASIP vs. **AMSR2** in [20%-80%] | 70 | 54.6, **25.8** | | | 30, **49** | | 36, **17** |
| grASIP vs. **AMSR2** in (80%-100%) | 79 | 88.4, **53.2** | | | 18, **37** | | 61, **42** |

**Table 3: Accuracy and matchup counts for data combinations specified. For the 20-80% SIC range calculation, data are inclusive (i.e. data greater than or equal to 20% and less than or equal to 80% are included). The 0-40% range and the 80-100% range are non-inclusive. 0-20% range is not considered, since only 11 datapoints fall in this range. Note that these accuracy estimates are a function of an imperfect input dataset, given the in situ observations do not cover all grid cells and all time points and are biased towards favorable operating conditions. grASIP data are underlined, AMSR2 data are in bold, and MASIE data are italicized.**

### 4.1.4 grASIP, AMSR2, and MASIE

grASIP's overall accuracy of 94.3% is high, but to understand what that value means, it is critical to compute the corresponding accuracy rate of other products as well. To do so, AMSR2 passive microwave data and MASIE synthesized data are considered, and the parity calculation is re-done. Recall that the temporal resolution of the three ice maps vary, and the SIC information varies, which imposes the following constraints on the in-situ data used for the accuracy calculation: (i) years covered, (ii) minimum SIC threshold, and (ii) binary ice vs. no ice information.

Given these constraints, since MASIE has an inherent cutoff threshold of 40%, a 40% SIC threshold must be used for grASIP, AMSR2, and the in situ observations. Therefore, for all four datasets (grASIP, AMSR2, MASIE, and in situ observations) when SIC < 40% it is considered as having no ice, while when SIC >= 40% it is considered as having ice. To do this calculation, we need in situ observations that provide SIC estaimtes, and thus exclude the binary in situ observations from this comparison.

The resultant overall accuracy estimates (Fig. 5 and Table 3) are similar to our original analysis for grASIP (Fig. 4), with grASIP exhibiting 90.5% accuracy, AMSR2 exhibiting 86.8% accuracy and MASIE exhibiting 90.0% accuracy, again with the caveat that the in situ data are not comprehensive and thus result in imperfect accuracy estimates. grASIP and MASIE tend to over-predict ice (n = 13 for grASIP and n = 12 for MASIE in Q3), whereas AMSR2 tends to underpredict ice (n= 23 in Q2). These results are not sensitive to the choice of cutoff SIC. Specifically, given the product accuracy of 5%-10% SIC, we repeat the calculation for a range of cutoff thresholds. The pattern is consistent at 40%, 45%, and 50% (grASIP and MASIE overpredict ice, AMSR2 underpredicts ice). At 35%, the pattern is true for grASIP (overpredicts ice) and AMSR2 (underpredicts ice), but MASIE is now even (overpredicts and underpredicts at the same rate). At 30%, grASIP is even, AMSR2 still underpredicts ice, and MASIE underpredicts ice.

Since most in situ observations in the subset of data queried for this calculation contain ice, these accuracy estimates favor products that over-predict ice (grASIP and MASIE). The weighted accuracy estimates (Table 2, calculation described in 3.1.1) reveal consistent accuracy estimates among all three products when this sample bias is taken into consideration. This overall similarity in accuracy estimates (90.5% vs. 86.8% vs. 90.0%; weighted accuracy 86.5% vs. 89.6% vs. 86.7%) was a surprise: given the dramatic differences in ice edge position (e.g. Fig. 1), one would expect similarly dramatic differences in accuracy.

The lack of such differences is explained in Section 4.2.

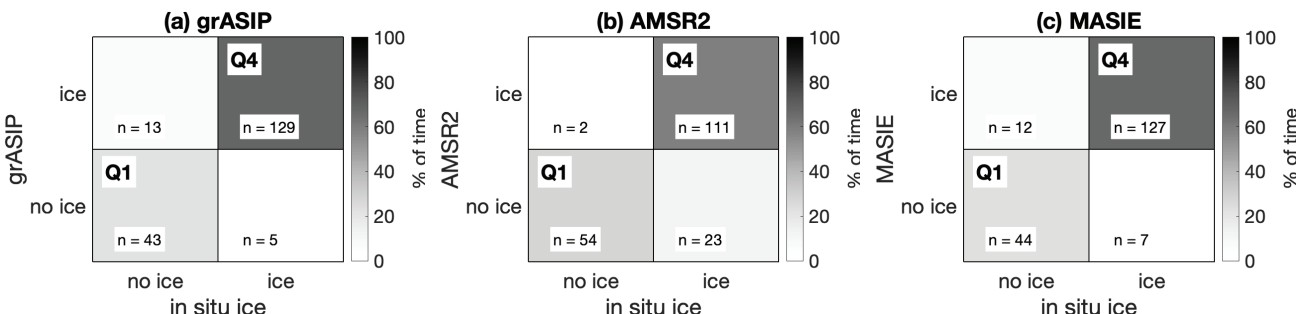

**Figure 5: Parity plot for the three ice products (grASIP, AMSR2, and MASIE) combined. In situ observations with ice concentration information (non-binary) from 2012-2022 are used. SIC cut-off is 40% for all four products (in situ observations, ASIP, AMSR2,**
**and by definition MASIE).**

### 4.1.2 grASIP vs. AMSR2

We now discard the SIC > 40% criterion for comparison of only grASIP and AMSR2 (but at this point keep the binary ice/no ice framework). The resulting accuracy is 95.2% for grASIP and 92.7% for AMSR2 (see Table 3 for details; weighted accuracies are 91.9% for grASIP and 85.0% for AMSR2). The primary difference between the datasets is an under-prediction
of ice by AMSR2 as compared to grASIP. Specifically, while grASIP incorrectly identifies 24 in situ observations as having no ice, when in fact ice is present, AMSR2 incorrectly identifies almost double that number, 51.

We now also discard the ice/no ice framework, in order to explore the accuracy of the products across all SIC ranges. We caution that there are high uncertainties embedded into this analysis for all three datasources: grASIP, AMSR2, and in situ
observations. Data are binned at 10% resolution, with the lower bound inclusive, except for at 0% and 100% (e.g. [0%-10%], [10%-20%),…[90%-100%]), similar to the methodology employed by Kern et al., 2019. This is consistent with the error bars on both the in situ values (estimated to be approximately 10%, Kern et al., 2019) and the error bars associated with converting concentration ranges to SIC values (see Fig. 2c,d, Table 1). As shown in Fig. 6, grASIP tends to overpredict ice and AMSR2 tends to underpredict ice, especially for SIC < 50%. This over- and under-prediction can be quantified by computing the root
mean squared error (RMSE) and the mean average distance (MAD) for the binned data and for the scattered data. In other

words, RMSE and MAD can be computed for the data that constitutes the shading in Fig. 6 (binned data), and the same statistics can be computed for the data that constitutes the scatter points in Fig. 6 (scattered data).

The resultant statistics indicate that AMSR2 under-predicts ice at a larger magnitude than that at which grASIP over-predicts
ice, both for the average data and for the binned data (Table 4). This is especially clear in the MAD binned data, where grASIP has a MAD of 3.6 (overprediction) and AMSR2 has a MAD of -5.4 (underprediction), but this pattern is consistent across the datasets. Therefore, while neither product is perfect, the over-prediction of ice by grASIP is smaller than the under-prediction of ice by AMSR2. Given the high uncertainties on each dataset (see section 3.2), more work is needed to confirm or disprove these results.It is also important to remember that grASIP data will never fall into the 10-20% SIC interval (see Fig. 6a, Table
1), which could impact these results as well.

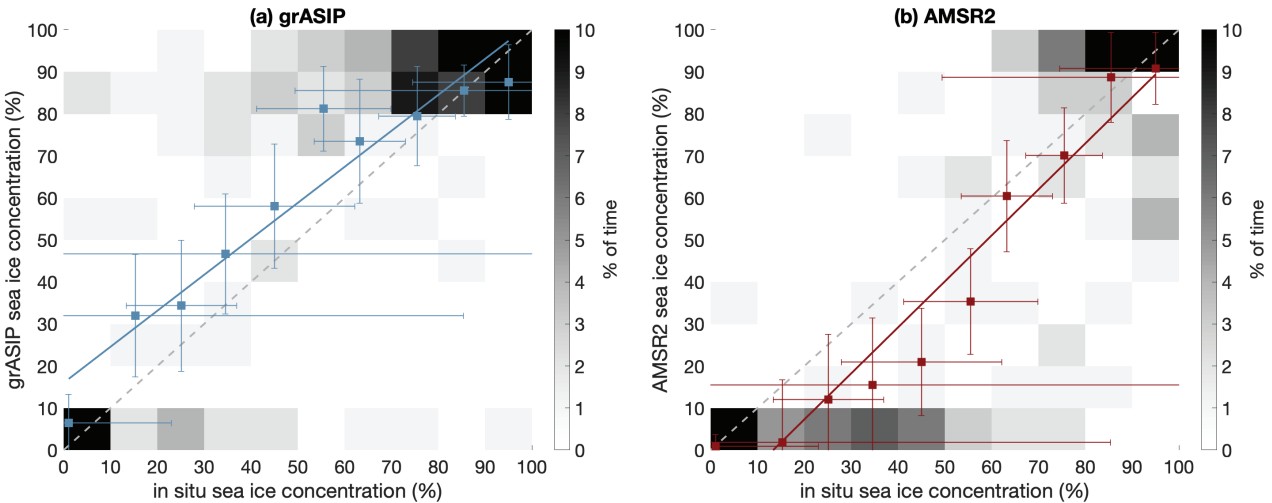

**Figure 6: SIC-based parity plot for grASIP and AMSR2. Red and blue squares indicate the product average for the given in situ concentration range (referred to as scattered data in the text), see text for details. Error bars indicate the standard deviation of the**
**values in that bin. Solid line indicates the linear fit through the product averages. Grey dashed line indicates the 1:1 line for reference.**

| | Scatter fit | | Binned data | | MIZ (20 – 80 %) | |
|---|---|---|---|---|---|---|
| # of datapoints | 10 | | 190 | | 81 | |
| | RMSE (%) | MAD (%) | RMSE (%) | MAD (%) | RMSE (%) | MAD (%) |
| grASIP | 12.4 | 8.9 | 21.8 | 3.6 | 27.2 | 11.6 |
| AMSR2 | 13.3 | -9.9 | 23.8 | -5.4 | 32.6 | -9.9 |

**Table 4: Root mean squared error (RMSE) and mean average difference (MAD) for the grASIP and AMSR2 SIC intercomparison. The first row indicates how many datapoints are included in the calculation. The first two columns represent RMSE and MAD for the scattered data (blue and red squares in Figs. 6a,b), the middle two columns represent RMSE and MAD for the individual binned**

**datapoints, rounded to the nearest tenth. The last two columns represent RMSE and MAD for the matchups in the MIZ, 20% – 80%, inclusive.**

### 4.1.5 grASIP vs. AMSR2 within the MIZ

As described in the introduction, we seek to assess the performance of different ice products in low-ice conditions. While MASIE reports binary ice information and cannot be used to isolate specific concentration ranges, grASIP and AMSR2 data

can identify the Marginal Ice Zone (MIZ), here defined to be between 20 and 80% ice concentration.

The resulting parity plot (Fig. 7) demonstrates that, although not perfect, grASIP outperforms AMSR2 in these low SIC environments. grASIP exhibits an accuracy of 55%, while AMSR2 exhibits an accuracy of 26% (a difference of 29%). Furthermore, if we do not limit the concentration reported by the product (in other words, we specify that the in situ

observations must be between 20 and 80%, but the products must simply either report or not report ice, regardless of the associated SIC value), these accuracy jump to 96% for grASIP (63 correct, 3 incorrect) and 76% for AMSR2 (50 correct, 16 incorrect), a difference of 20%.

However, this analysis is limited by the availability of in situ observations within the MIZ. Recall Fig. 3d, which illustrates

that the majority of in situ observations are either in open water or in dense ice. Additionally, recall the strikingly similar accuracy estimates from the parity calculations presented in Fig. 5. Both results can be explained by considering two case studies.

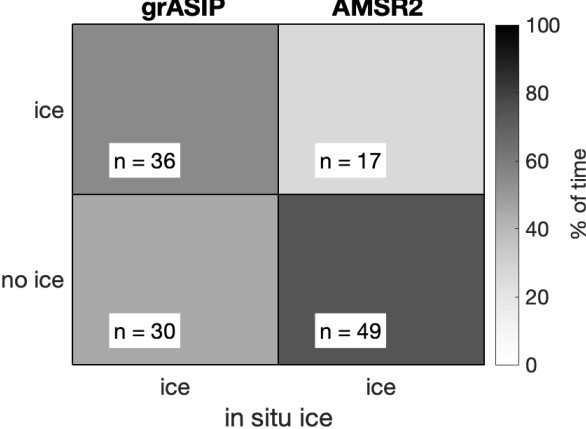

**Figure 7: Parity plot for grASIP and AMSR2 within the MIZ. In situ observations with ice concentration information (non-binary) from 2012-2022 are used. Concentration cut-offs are 20-80%, inclusive. Note the different horizontal scale used here, relative to previous parity plots: we are testing only within the MIZ, so only where in situ observations show ice.**

### 4.2 Case studies

Consider the position of the in situ asset on October 2, 2015 (Fig. 8a). The ship is far north within the pack ice, reporting a compact ice cover, and far from the ice edge regardless of which ice edge is considered. In this case, all products agree that

the asset is in the ice. However, all three products disagree on the position of the ice edge (regardless of whether the ice edge is defined as 20% or the MASIE definition of 40%). In this case study, the grASIP ice edge is further south than both the AMSR2 and MASIE ice edges, especially east of 158°W. The MASIE ice edge is similar to AMSR2 west of 152°W, but east of this longitude the MASIE ice edge is between the grASIP and AMSR2 edges. Similarly, consider the position of the in situ asset on September 7, 2017 (Fig. 8b). The ship is far south of the ice edge, reporting open water conditions. All products agree

that the ship is in open water, despite all three products reporting different ice edges. In this case study, the grASIP ice edge is systematically further south than both the AMSR2 and MASIE ice edges throughout the domain; AMSR2 and MASIE have relatively similar ice edge positions. In these two examples, while the ice edges vary dramatically between products, the in situ assets are sampling in geographic positions and SIC ranges where the products are more likely to agree.

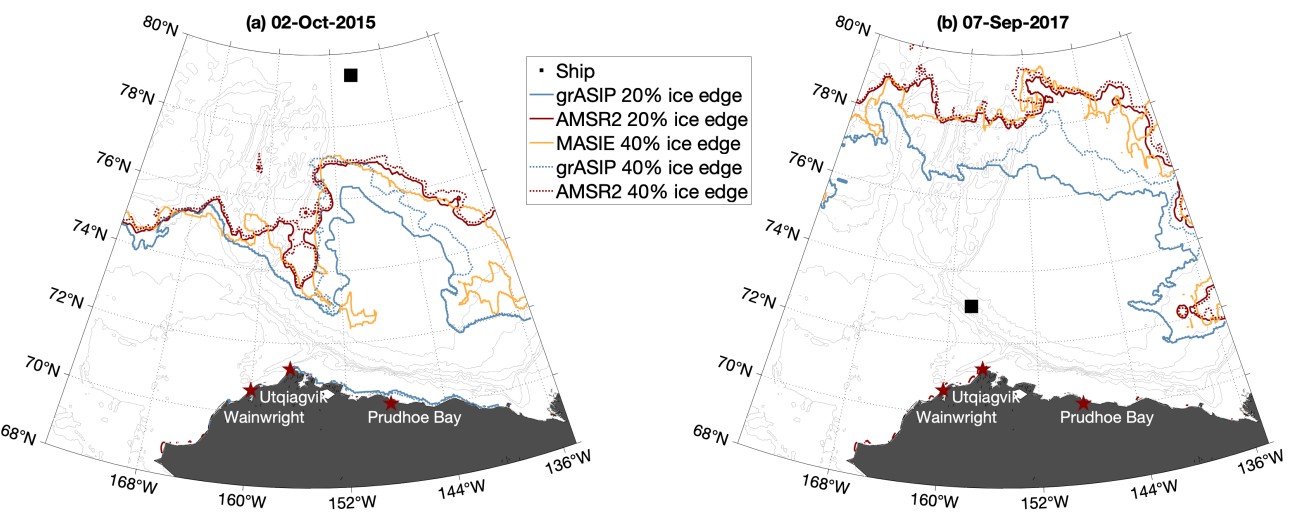


**Figure 8: Two case studies of ice edge position for all three products, with both the 20% and 40% ice concentration contour plotted from grASIP and AMSR2. Black square indicates the location of the in situ observation.**

Recall the distribution of in situ assets as a function of ice concentration (Fig. 3d), which shows that in situ assets are

predominantly found in either open water or high SIC. Thus, the in situ observational database has a poor sampling of the MIZ, which is unfortunately exactly the region where the products most strongly disagree (Fig. 7, see Table 4).

**4.3 Ice edge comparison**

Here we present a comparison of ice edge position between satellite products; specifically, the average distance between ice edges in two different products is computed. While we are unaware of any studies that calculate the average distance between

ice edges in different products, finding the distance between two contours is a common calculation used to define the width of the MIZ. There are a variety of algorithms implemented to compute the distance between two contours. For example, Strong

et al. (2017) compared four methodologies and concluded that a streamline definition solving Laplace's equation is the most rigorous approach, albeit computationally intensive. Strong (2012) and Strong and Rigor (2013) rely on this methodology to describe the impact of atmospheric conditions on the Atlantic Arctic MIZ and on long-term seasonal trends in the MIZ, respectively. Alternatively, Stroeve et al. (2016) use a radial method to compare the detection of the MIZ and consolidated pack ice and coastal polynyas in two passive microwave algorithms. Using this methodology, radial sections are traced; the algorithm then flags the first point along each radial section where ice exceeds a minimum threshold (e.g. 15% SIC) and then continues along the section until the ice reaches a maximum threshold (e.g. 80% SIC). Here, we follow the radial methodology, for its simplified computational demands and intuitive frame of reference, and we adapt the methodology to this study as follows.

Here, the calculation is done by first gridding AMSR2 and MASIE onto a 0.05° grid to match that of grASIP, and second by creating a land mask which is extended seaward by five pixels in each direction to remove instances of land spill over (e.g. Cavalieri et al., 1999). Additionally, masks are implemented over Wrangel and St. Lawrence Islands, and a two-dimensional 3 x 3 pixel rectangular running mean is applied to remove small-scale features and to facilitate a large-scale analysis of basin-wide ice edge position. Then, for each longitude (at 0.5° resolution), the latitude of the ice edge is calculated (either 20% or 40%), as defined by the first gridcell that SIC exceeds 20% or 40%, for each longitude and for each day.

This permits a comparison between ice edge location at each longitude between product pairs. For each day and for each longitude bin, the north-south distance between the two products is computed. For example, if the 20% contour in grASIP and AMSR2 is being compared, the distance between the grASIP and AMSR2 20% contour at each longitude is computed daily. The longitudinal mean is then calculated for each day, which results in a time series of ice edge distance between products as a function of day.

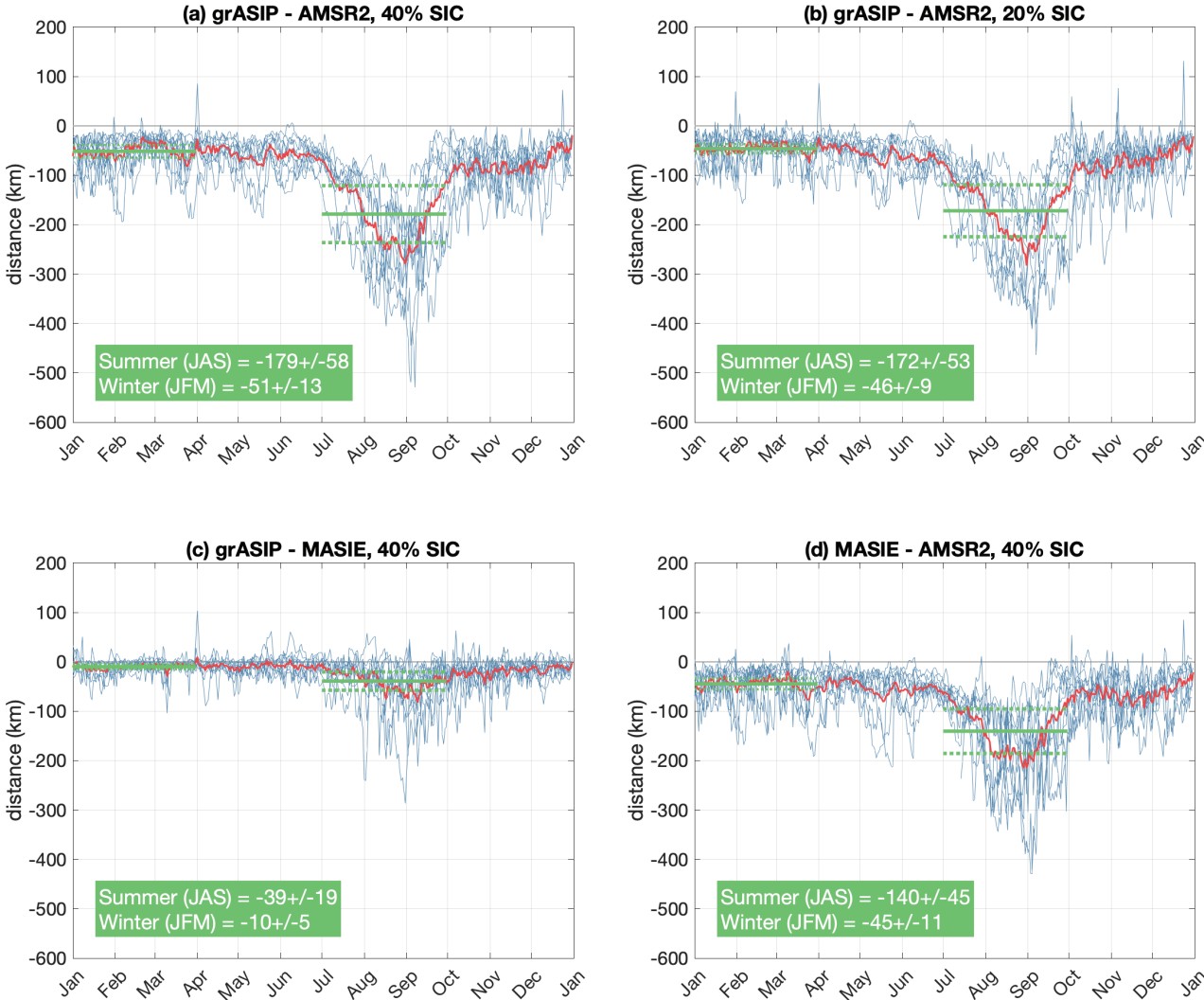

**Figure 9: Distance between ice edges.** (a) Longitudinal-mean distance between the grASIP and AMSR2 40% concentration contours. Blue contours represent yearly timeseries, red contour represents the mean of all years (2013-2022). (b) Same as (a) but for grASIP and AMSR2 at 20%. (c) Same as (a) but for grASIP at 40% vs. MASIE. (d) Same as (a) but for MASIE and AMSR2 at 40%. Negative values indicate that the first product (e.g. grASIP in panels a-c and MASIE in panel d) is further south than the second product. Green solid lines indicate seasonal mean (January, February, March for winter; July, August, September for summer); green dashed lines are the standard deviation envelopes.

The average distance between the grASIP and AMSR2 ice edges exhibits distinct seasonal variability, with grASIP an average of 172 (179) km south of AMSR2 in summer and 46 (51) km farther south in winter for the 20% (40%) ice edges. The comparison between grASIP at 40% and MASIE demonstrates the same pattern, but with significantly reduced distances: the

grASIP ice edge is 39 km farther south in summer and 10 km farther south in winter. Finally, the MASIE ice edge is generally farther south relative to the AMSR2 ice edge in summer (140 km), and is 45 km south of the AMSR2 ice edge in winter.

    This calculation highlights that in all cases and in all seasons, the grASIP ice edge is further south than the ice edge in AMSR2 and MASIE. Furthermore, the distance between ice edges is greater in summer than in winter in all four intercomparisons.
This is likely due to the presence of melt ponds and snow and melt on the surface of the ice, as passive microwave tends to struggle when water from melt is present on the surface of the ice during the summer season (e.g. Kern et al., 2020, Cavalieri et al., 1990).

    These systematic differences have been recognized in previous studies; Meier et al. (2015) noted that MASIE exhibits a larger
sea ice extent in summer months as compared to passive microwave measurements (similar to the result shown in Fig 9d, but for area instead of distance). As with the Meier et al. (2015) passive microwave and MASIE comparison, all three datasets analyzed here include inherent sources of error that can help put the systematic differences in Fig. 9 into context. For AMSR2, as previously described, ice is under-estimated because of wet snow and melt ponds on the surface of the ice and because of the presence of thin ice. For MASIE and ASIP, potential sources of error include the lack of high-resolution imagery on any
given day, if conditions are cloudy. This could result in an analyst being unwilling to move the ice edge until another clear image is obtained, which results in an ice edge that looks to be loitering but is in fact simply a replica of the previous day's ice edge (e.g. Meier et al., 2015). The lack of high-resolution imagery could also result in a reliance on passive microwave data to draw the ice edge; while we cannot comment on the frequency with which analysts rely on passive microwave measurements alone, Meier et al. (2015) present an example from September 2010 where the MASIE ice edge remains fixed until the ice
quickly retreats over the course of one day, citing this as an example of an analyst likely waiting for a clear image. A similar phenomenon was documentd by Steele and Ermold (2015), who defined a loitering ice edge as an ice edge that remains in a fixed geographic region for multiple days in a row. They showed that MASIE exhibited a higher tendency for loitering when compared to passive microwave data. This suggests it is not common for analysts to rely solely on passive microwave measurements, and instead suggests that analysist instead opt to wait for new imagery before moving the ice edge.
Additionally, ASIP uses polygons that denote sea ice concentration ranges; the polygon boundaries, and the ice concentration ranges delineated by these polygons, are not overly well-defined and repeated analysis could result in a slightly different designation of polygon locations and concentration ranges. Finally, one might ask whether ASIP's primary task is to prioritize navigational safety and thus analysts can be conservative in their ice estimates. ASIP ice analysts do not have this directive. Of course, implicit bias could result in a tendency to overestimate the true ice conditions if an analyst errs on the side of
caution, but this is not a stated edict at ASIP.

## 5 Discussion

In this study, we show that grASIP generally has an ice edge further south than MASIE and AMSR2, and that it is generally more accurate, and exhibitis more fine-scale structure, when compared with shipboard observations. Thus, if a scientist or operational stakeholder needs to know how likely it is that ice of any SIC is present at a location in the Alaskan Arctic, then

grASIP is the best choice product to use to address this question.

In most of this study, a presence/absence of ice framework was used; however, when the SIC values are retained, there is evidence that grASIP performs better than AMSR2 in most concentration ranges and especially within the MIZ (Fig. 7). At low SIC, both grASIP and AMSR2 show high estimated SIC variance, with grASIP systematically over-predicting SIC and AMSR2 under-predicting SIC (Fig. 6). However, the under-prediction in AMSR2 is larger the over-prediction in grASIP,

illustrating that while neither product is perfect, grASIP outperforms AMSR2. That said, in situ observations of the MIZ are lacking (see Fig. 3d), indicating that there is a clear need for measurements of SIC in this subdomain. We hypothesize that more in situ observations of the MIZ might in fact change the accuracy statistics presented here.

### 5.1 Limitations in the datasets

*In situ observations:* Matchups within the MIZ can be quite problematic, for a variety of reasons. Despite the 5-10% error bars

presented by Worby and Cosimo (2004) for the Ice Watch in situ observations, it is important to recall that this is for a 1 nautical mile radius around a ship, dependent on both the observer and the visibility at that time. Of course, the ship could be moving during the 10-minute sampling window, thus elongating the area assessed by this in situ observation (e.g. Bietsch et al., 2015; Kern et al., 2019). Additionally, weather and visibility can impact the observable radius around the ship (e.g. Kern et al., 2019). This surveyed region is only a small portion of a satellite grid cell, which makes matching up the ship-based

estimate with satellite estimates challenging. This is especially true in low SIC environments, where small-scale ice floes and features dominate the grid cell. As such, if a ship is moored to a floe (which is often the case during on-ice buoy deployments or long-term drifts), it will sample heavier ice conditions than are reflected in the broader grid cell. Conversely, most ships will preferentially navigate through leads or low ice regions, potentially biasing the in situ observations to low values. (e.g. Kern et al., 2019). Furthermore, taking daily averages of ship position and SIC complicate this calculation, as a ship is more

likely to move across a large range of SIC measurements in one day when in the MIZ than when in heavy ice, because a ship can move faster in low SIC conditions than when backing and ramming in high SIC environments. As such, the ship may transit from 15% to 50% to 100% ice in one day, and these values would average to 55% ice cover, despite the fact that at the time of satellite passage, the ship may be closer to the low or high ends of this range.

*ASIP:* As described previously, ASIP (and most analyst-generated blended SIC maps) are particularly valuable given the

high spatial resolution afforded by the trained attention of an analyst and the data representation as SIC polygons. However, ASIP inputs have changed over time as satellite imagery has improved, the human analysts responsible for the SIC maps

have changed over time, and often, bad weather means that imagery may not be available for a specific day or series of days. Therefore, ASIP is better suited to understanding the daily state of the ice pack, rather than being used to compute long-term climate trends.

Furthermore, ASIP data are presented using specific SIC ranges, which results in twelve SIC values present in the gridded maps (e.g. Table 1). As such, these maps do not provide continuous fields in space and time. In space, this is true because adjacent polygons can be described with a different SIC range, thus resulting in a discrete jump in SIC value when moving from one gridcell to another, or from one polygon to another (e.g. Fig 2b,e). In time, the same is true; if a polygon is described as having 8/10 – 10/10 SIC on one day, and then a similar polygon in a similar region is described as having 6/10

– 8/10 SIC the next day, a timeseries of a gridcell within both polygons will exhibit a discrete SIC change from 90% to 70% on a given day. This is common for gridded maps originating in shapefile-based SIC maps (e.g. USNIC, 2020). This motivates the creation of spatially smoothed SIC maps in space and time from the ASIP source files, which is ongoing.

*AMSR2:* While AMSR2 data provide a valuable long-term climate record with smooth SIC fields from 0-100%, passive microwave measurements struggle to capture SIC conditions when water from melt is present on the ice surface (e.g. Kern et

al., 2020, Cavalieri et al., 1990). Furthermore, AMSR2 at 3.125 km resolution is not simply data at this resolution, rather this spatial resolution represents a blend of frequency channels, each with their own associated footprint (see Section 2.3.1 and Melsheimer (2024) for more details).

*MASIE:* Like ASIP, MASIE maps originate with a human analyst performing a synthesis of available ice imagery. As such, changes to the analysts responsible for map generation, changes to satellite technology and imagery capabilities, and weather

conditions can impact the daily SIC maps and renders MASIE maps useful for understanding the daily state of the ice pack, rather than long-term climate trends.

Despite these limitations, it is evident that more in- situ observations of the MIZ are needed and would likely impact the results of our parity analysis. Further, additional validation of data products using SAR imagery might be useful. The results of this study, both grASIP validation and the intercomparison between grASIP and AMSR2 and MASIE, indicate that grASIP is a

valuable product to include in scientific analysis of ice conditions, especially in low SIC environments, during periods of active melt, and when isolating a high-resolution ice edge. We note that our gridded grASIP SIC product provides a relatively accurate field (compared to in situ observations) and thus can be used as an optimal "state estimate" of SIC in the Pacific Arctic on any given day. This should be useful for a variety of scientific studies, including numerical model validation.

## Competing Interests

The authors declare that they have no conflict of interest.

## Author contribution

AP and MS formulated the study. AP wrote the code, performed the analysis, and drafted the manuscript. All authors advised on the methodology and the interpretation of findings. All authors reviewed and edited the manuscript.

## Acknowledgements

The authors thank the captains, crews, and observers onboard all 23 cruises utilized in this analysis. The authors thank the NOAA Saildrone mission teams from 2019 and 2022 for successful missions, especially Andrew Chiodi and Marisol García-Reyes for providing access to the data. We are grateful to all ice analysts at ASIP and USNIC who contributed to the ice maps over the years, and we acknowledge Harry Stern, Melinda Webster, Axel Schweiger, Severine Fournier, Odilon Houndegnonto, and Jennifer Hutchings for useful discussions. We thank Florence Fetterer and one anonymous reviewer whose comments strengthened the manuscript. A.P. was funded by NSF OPP-2219147. M.S. was funded by NSF OPP-1751363, ONR N00014-21-1-2868, and NASA 80NSSC18K0837 and 80NSSC21K0832.

## Code Availability

The codes used to read, reformat, project, and grid the National Weather Service Alaska Sea Ice Program data are available on GitHub at https://github.com/astridpacini/NWS_ASIP. The Climate Data Toolbox for MATLAB (Greene et al., 2019) was utilized to compute land masks.

## Data Availability

The AMSR2 data at 3.125 km resolution were downloaded from the University of Bremen at https://data.seaice.uni-bremen.de/amsr2/asi_daygrid_swath/n3125/. AMSR2 at 25 km resolution were downloaded from the National Snow and Ice Data Center at https://nsidc.org/data/au_si25/versions/1. MASIE data are also available for download from the National Snow and Ice Data Center at https://nsidc.org/data/g02186/versions/1#anchor-1. ASIP ice charts were provided by ASIP analysts and are available upon request via email at nws.ar.ice@noaa.gov. All Ice Watch data were downloaded from the Ice Watch data repository at https://icewatch.met.no/about. The two DBO cruises utilized in this study were obtained from the Arctic Data Center (CCGS Sir Wilfed Laurier 2016 cruise: https://arcticdata.io/catalog/view/doi%3A10.18739%2FA27P8TD2J; USCGC Healy 2017 cruise: https://arcticdata.io/catalog/view/doi%3A10.18739%2FA25Q4RM2M). The data from the Study of Under Ice Blooms in the Chukchi Ecosystem USCGC Healy 2014 cruise were obtained from the Arctic Data Center at https://arcticdata.io/catalog/view/doi%3A10.18739%2FA2416T03D.

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
