# Peer review of "National Weather Service Alaska Sea Ice Program: Gridded ice concentration maps for the Alaskan Arctic"

_EGUsphere, 2024_

## Author Comment (AC1)

Review of National Weather Service Alaska Sea Ice Program: Gridded ice concentration maps for the Alaskan Arctic by Pacini, A., et al.

Summary:

This manuscript introduces gridded sea ice concentration maps available since 2007 from the National Weather Service Alaska Sea Ice Program for the Alaskan Arctic. The main content of the manuscript is the evaluation of this new product (called ASIP henceforth) by means of comparing it with independent data. These are ship-based observations and saildrone images, the MASIE ice extent product and a high-resolution sea ice concentration product. The comparison shown focusses a lot onto so-called parity plots in which the hit and false alarm rates of binary ice information provided and/or derived from the products is compared with each other. Conclusions are drawn from these plots; in addition to these the authors also look a bit into the comparison of the actual sea ice concentration values and take a look at the location o the ice edge and how this intercompares between the different products usd.

I am listing a number of general concerns first. Subsequently you find a number of specific comments which in part detail the general concerns further. I also have a few editoral comments / typos.

We thank the reviewer for the thoughtful comments and suggestions, which have strengthened the manuscript. Please find our detailed responses in blue below.

General Comments:

GC1: I have a major concern with the scientific rationale and motivation to evaluate a product (your product) providing more information than just binary ice / no ice mainly by means of reducing the information content to compare it with evaluation data that (only!) partly also come as binary information. This I really don't understand and find it neither convincingly explained in the manuscript nor do I find compelling evidence in the manuscript that doing the evaluation this way really adds value and provides credible and useful results.

The choice to perform this analysis using a binary ice/no ice framework was made for two main reasons, both underscored by the importance of not propagating large error bars that are present in the datasets utilized. The specific reasons for this choice are explained below and clarified in the text in a new section within the methods section (3.2 Motivation for and conversion to binary ice/no ice).

As written in the text: With the exception of section 4.1.2, our comparisons between ice maps and in situ observations are performed in this binary ice/no ice framework, which we feel is best for two main reasons. First, human observations of SIC are subjective. While Worby and Comiso (2004) quote an uncertainty of 5% - 10% on human-made SIC observations, this is likely an underestimate. As Kern et al. (2019) describe, untrained or less experienced observers will be able to assess SIC at low concentrations (<30%) and at high concentrations (> 80%) relatively easily; however, human observers will naturally struggle in the SIC ranges between these extremes, which is a region of particular interest in this study. Even among experienced

observers, there will be discrepancies in the identified SIC exceeding 10%. Further complication and error is introduced because of constraints on visibility, speed of the ship, distribution of floes and preferential navigation to avoid ice, or to raft to an ice floe, and more. Our second reason for choosing a binary framework has to do with the ranges of SIC provided by ASIP, which are often order 20%. This means that a grid cell reporting 80% could actually exhibit 70% or 90% SIC and still fall in that particular classification range. Additionally, ASIP polygons and ranges are assigned by human observers with variable input data, and thus subjectivity is introduced in the designation of such polygons. When coupled, these two reasons can result in an error range of 40%, almost half of the SIC range.

GC2: The manuscript is not convincing with respect to the description of the steps that are undertaken to i) grid all data into one common grid and to ii) explain how data sets are reduced in their information content from sea ice concentration to binary information - including the assciated uncertainty that is involved in this conversion process.

We have updated the text to provide more details on the gridding process, the conversion to binary information, and the associated uncertainty of this conversion in the new Methods section, and in the Data section.

GC3: I am not convinced that the suite of parity plots that is presented are the optimal solution to show the quality of the new data set that you are evaluating in your manuscript. While I believe 1-2 specific parity plots could stay - especially when these are used to compare data that are per se binary, i.e. the saildrone data and MASIE, I very much recommend to work more with 2-dimensional histograms such as the one shown in Figure 6 and work along the lines of computing mean and median differences (also the absolute ones) and their standard deviations. This appears to me a more quantitative way to evaluate the ASIP product in its current form.

As described in the response to GC1, we choose to focus on binary comparisons to be as cautious as possible, given the large uncertainties on percentage-based SIC values. Therefore, we have retained the parity plots but have modified the plots according to the suggestions in GC4. Furthermore, the statistics suggested by the reviewer (e.g. RMSE, mean average difference) are now included in Section 4.1.2.

GC4: In case the parity plots stay as a central element of the manuscript I recommend to reshape them such that they use the space given in the manuscript more efficiently - i.e. decrease the block size but increase the font size.

We have reshaped the parity plots according to the reviewer's suggestions. We have decreased the block sizes and increased the font sizes.

GC5: The Discussion section should be before the Summary section. The discussion section should furthermore discuss in substantially more depth the limitations of the data sets involved - as laid out in my respective specific comment.

We have rearranged the Discussion section as follows: Within the Discussion section, we have added a subsection 5.1 Limitations in the datasets, which is then broken down by the four main

datasets used in this manuscript (in-situ observations, ASIP, AMSR2, and MASIE) and the challenges associated with each dataset are presented.

GC6: I find room for improvement in the structure of the manuscript. I find that data, methodology and results are in part quite mixed and call for a better organization in that respect.

The manuscript has been restructured to have separate Data and Methods sections. The Methods section now details the Parity calculation (3.1), the motivation for and conversion to binary ice/no ice (3.2), how the ice edge is defined (3.3), and acknowledgement of footprint size variations among datasets (3.4). Furthermore, we have carefully edited the manuscript in line with the reviewer's suggestions to move discussions of data to the Data section.

Specific Comments:

L19: "in-situ asset distribution" --> Not immediately clear what you mean with this. What do you mean by "asset" in this context?

We have updated the text to state "in situ observations' geographical distribution".

L56: I suggest to add at least Lavergne et al., 2019, https://doi.org/10.5194/tc-13-49-2019 to this list since it adds a novel approach. Also, you might want to point towards the Ivanova et al., 2014 10.1109/TGRS.2014.2310136 / 2015 doi:10.5194/tc-9-1797-2015 papers here since these provide a good overview of the different existing approaches.

Thank you for these suggestions. We have included the studies in the text and provided a brief explanation of Ivanova et al. 2014 and 2015. We note that Ivanova et al., 2015 was already cited in that section, but we have elaborated on their analysis.

L64: "synthetic aperture radar" --> There is a growing number of sea ice cover / sea ice concentration products based on SAR data; recent years have seen a boost in such maps thanks to more frequent coverage of the polar regions with SAR images and advanced computational tools. I was wondering whether you should not come up with a few examples of such tools / products for completeness. There is for instance the "MAGIC" tool (see Leigh et al., 2014, 10.1109/TGRS.2013.2290231 ) and there are other products, e.g. DTU_AI4Arctic.

We have updated the text to include discussion of specific SAR-based products.

L68-70: "Operational ... products." --> I agree only partly to this statement because operational ice charts - at least those of most ice services - use polygons to provide information of groups of dominant ice classes. In addition these only provide ice concentration ranges of, e.g., 10% resolution.

While it is true that operational ice charts provide polygons, we note that these polygons can be smaller than the footprint of some satellites. Furthermore, the text indicates that these operational charts can be higher resolution than passive microwave counterparts, not that they always are.

The text has been updated to clarify this. The point about ice concentration ranges is noted, and we have updated the manuscript to clarify this.

L76: You refer to "many ... techniques used" but you do not further refer to them. Is this on purpose? Because, in what follows you rather report on the results of evaluation studies dealing with two such different products. And in contrast to the CIS sea ice charts the MASIE product is not an operational sea ice product that can be used for navigation but is simply another form of deriving the sea ice extent. I was therefore wondering whether first mentioning a few more "real" ice charts, such as from NIC, AARI, and the Norwegian, Danish and Finish ice services would not make sense.

Thank you for this point. We have updated the text to include a discussion detailing these ice services, including the USNIC, the Danish Meteorological Institute, the Norwegian Meteoroloigical Institute, the Finish Meteorological Institute, the Arctic and Antarctic Research Institute, and the Canadian Ice Service.

I note that it would be helpful to provide the period (i.e. number of years) that were used in the two evaluation studies mentioned.

Done.

L82/83: It might make sense to emphasize that this larger sea ice extent reported for MASIE by Meier et al. (2015) is particularly large / pronounced during summer melt, right?

This is only partially correct; Meier et al. (2015) found that the MASIE sea ice extent exceeded that of passive microwave throughout the year, except for during in May/June and again at the end of melt season/start of freeze-up (Sept/Oct) (see their Figs 2 and 3a). Therefore, we leave the text as-is.

L147-153: Have these maps ever been compared to AARI or NIC charts? If not why not?

ASIP and the NIC use similar methods to generate ice maps and collaborate closely; however, they have independent data streams. To our knowledge, these datasets have not been compared in the scientific literature. This has been added to the text.

L175-177: "Polygons ... a larger polygon." --> This I don't understand ... Does this mean that if there is a large polygon containing 70-90% sea ice concentration within which there is a smaller polygon with 10-30% sea ice concentration will result in the entire area (small + large polygon) to be displayed as 10-30% sea ice concentration? Please modify your writing such that it becomes more clear.

Thank you for pointing out this confusing wording. No, this means that for the spatial extent of the smaller, embedded polygon, the value of that small polygon is used in the gridded polygon. For the area outside the small polygon, the large polygon value is used. The text has been updated to clarify this.

Table 1: There is no SIC value in the last row. Does this mean that a value of 100% is never given - also not for landfast sea ice? This reads a bit strange I have to admit.

Thank you for pointing this out. The last row should have 100% SIC for landfast ice. The table has been updated accordingly.

249-252: While details of the respective data analysis can be found in the Chiodi et al paper I would like to see a more balanced approach (when compared to the ship-based observations) and ask for some basic description about the spatial and temporal resolution of these saildrone data, the observations height and approximate "footprint" and information like this.

We have provided further details on Saildrone imagery, including resolution, height, and footprint in the data section and in the new subsection within the Methods section on footprint size (section 3.4).

L223/224: Worby and Comiso (2004) studied Antarctic sea ice and hence "evaluated" the ASPeCt observations; ASSIST is something which was combined with ASPeCt substantially later, kind of in parallel to the ASPeCt / ASSIST data set that is available, e.g. here: https://www.cen.uni-hamburg.de/en/icdc/data/cryosphere/seaiceparameter-shipobs.html

This point is well-taken; we have updated the manuscript to reflect the distinction between ASPeCT and ASSIST/Ice Watch.

L239/240: How is this conversion done? Please give a description here or refer to the place in the paper where the respective information is given.

Done. We now include a section within the Methods section dedicated to "Motivation for and conversion to binary ice/no ice" (Section 3.2).

L241/242: I am not sure the mentioned "subjectivity" is something you need to remove - for two reasons. First of all, also the ice charts contain a certain degree of subjectivity. Secondly, the ASPeCt / ASSIST sea ice observations have a reported uncertainty which is similar to the one you reported in the previous section about the ASIP data set; it is around 5-10%. So the uncertainties are the same and I do not see added value to assess ASIP with binary ice/no-ice values.

See responses to GC1 and GC3.

L260-263: "It utilizes ... stereographic grids" --> This needs to be rewritten. The framing information is:

Thank you for this clarification. We have clarified the text and expanded on the details noted below.

- AMSR2 is a multi-frequency passive microwave sennsor that provided brightness temperatures at a number of different frequencies; one of these is 89GHz.

This is now explained in the text (section 2.3.1).

- The ARTIST algorithm has been developed for SSM/I data (Kaleschke et al., 2001), adopted to AMSR-E data (Spreen et al., 2008) and then applied to AMSR2 data - without further tie point modification as far as I know.

This is now detailed in the text. Tie points are only used when the ASI algorithm is used to convert from brightness temperatures to SIC at 0% and 100% and have not been modified from the AMSR-E tie points (e.g. Beitsch et al., 2014).

Beitsch, A.; Kaleschke, L.; Kern, S. Investigating High-Resolution AMSR2 Sea Ice Concentrations during the February 2013 Fracture Event in the Beaufort Sea. *Remote Sens.* **2014**, *6*, 3841-3856. https://doi.org/10.3390/rs6053841

- Sea ice concentration data are derived using the brightness temperature polarization difference of the 89 GHz channels (not from "swath brightness data").

This is now included in the text.

- I invite you to check whether the brightness temperatures aren't first gridded into the polarstereographic grid before the SIC is computed. You might want to check the documentation.

Based on the documentation, gridding is not performed until after SIC is computed. The swath data are converted from brightness temperatures to SIC, and then the gridding is done for daily swath data.

Documentation: https://data.seaice.uni-bremen.de/amsr2/ASIuserguide.pdf

And as a comment: You use this ASI algorithm SIC data for kind of an "evaluation". While this is of course fine I was wondering whether you can report about any validation studies that report about the accuracy of the AMSR2 SIC product provided by the University of Bremen. How reliable is this data set? It is credible to use this data for an evaluation?

We have now included a discussion of the error estimates on these data, including a brief discussion of studies that have evaluated both the ASI algorithm, and the 3.125 km resolution product, against a variety of data sources to show that these are credible data for the evaluation in this study. This can be found in Section 2.3.1, paragraph 2.

L275: So you regrid the MASIE data but you do not regrid the AMSR2 SIC data? At least you did not comment on that in the previous paragraph.

Thank you for catching this. Both AMSR2 and MASIE are re-gridded for the ice edge intercomparison analysis (section 4.3). The text has been updated to reflect this.

L280-282: Again my question why? Why did you not use the concentration values as provided?

And: How did you do the ice / no ice conversion for these data sets?

Please see the responses to GC1 and GC3.

We have elaborated on the creation of this binary ice/no ice logical for each dataset in the new Methods section 3.1 (Parity analysis) and on the rationale behind a binary ice/no ice framework in the new Methods section 3.2 (Motivation for and conversion to binary ice/no ice).

L294/295: While it is true that historically 15% has been used as the SIC threshold to define where there is ice, I find your approach not well motivated. ASIP provides non-binary observations (see Table 1) and these should be evaluated - not the binary value.

Please see the responses to GC1 and GC3.

We thank the reviewer for raising this concern, as it highlighted that ASIP (now referred to as grASIP based on a comment from reviewer 2) will never report SIC of 15% (Table 1). As such, we re-compute all statistics for a cutoff SIC of 20%: the results are not qualitatively or quantitiatively different than when the 15% SIC threshold is used.

In addition, seeing that you included MASIE which uses a 40% threshold to define between ice and no-ice, I get confused about the credibility of your results. This does not look like a well-thought through intercomparison approach, I am sorry.

We note that in this section, grASIP is the only data product considered. Later, when MASIE is considered, the 40% threshold is used to delineate between ice/no ice among all four products (grASIP, AMSR2, MASIE, and in situ observations), not just MASIE, to ensure a credible and appropriate intercomparison.

L296-298: "We note ... with ice" --> I don't understand this sentence.

This means that Saildrone camera images do not give us sea ice concentration, only presence of absence of ice. Here we assume that anytime the Saildrone camera sees ice, then SIC>15% (see Fig 1 of Chiodi et al., 2019). This has been clarified in the text.

L313-319 ... what is given here is essentially a description of the methodology. I suggest to have a more clear structure in the paper, with a Data section, a Methods section and then a Results section.

We have created a new section for Methods, which includes subsections for 3.1 Parity analysis, 3.2 Motivation for and conversion to binary ice/no ice, 3.3 Defining the ice edge, and 3.4 Footprint size. This text explaining the three-way parity calculation has been moved to the Methods section, under 3.1 Parity analysis.

L319-323: This information actually belongs to the section where you described the saildrone observations / data.

This information has been moved to the Data section where the Saildrone data are described (2.2.2 Saildrones).

L327/328: I don't understand how data products (e.g. ASIP or MASIE) can "report" an accuracy. Please re-consider your writing.

We have changed the wording throughout the manuscript to say that data products exhibit a specific accuracy, or that we estimate an accuracy.

L328/330: I don't think it is a credible approach to refer to over- or under-prediction of ice when the respective SIC ranges that you are considering here are as large as 40% or 60%.

What happens to these n=13 or n=12 (Q3) and n=23 (Q2) values if you would change the threshold value of 40% used by 5% or 10%, i.e. the uncertainty of the involved products?

The results are not sensitive to the choice of cutoff threshold. See table R.1 below, which repeats the parity calculation for a cutoff threshold of 30%, 35%, 40%, 45%, and 50%. The pattern is consistent at 40%, 45%, and 50% (ASIP and MASI overpredict ice, AMSR2 underpredicts ice). At 35%, the pattern is true for ASIP (overpredicts ice) and AMSR2 (underpredicts ice), but MASIE is now even (overpredicts and underpredicts at the same rate). At 30%, ASIP now over and under predicts at an even rate, AMSR2 still underpredicts ice, and MASIE underpredicts ice. This is now explained in the text.

|  | Q1 | Q2 | Q3 | Q4 |
|---|---|---|---|---|
| **30%** | 39, 45, 41 | 8, 27, 10 | 8, 2, 6 | 135, 116, 133 |
| **35%** | 41, 50, 43 | 7, 23, 8 | 10, 1, 8 | 132, 116, 131 |
| **40%** | 43, 54, 44 | 5, 23, 7 | 13, 2, 12 | 129, 111, 127 |
| **45%** | 45, 57, 46 | 6, 21, 5 | 14, 2, 13 | 125, 110, 126 |
| **50%** | 47, 62, 47 | 4, 17, 4 | 19, 4, 19 | 120, 107, 120 |

R.1 Matchup counts in the four quadrants, for a range of cutoff thresholds. grASIP is blue, AMSR2 is red, MASIE is yellow.

Figure 5:

- The font size used is quite small.

Fixed.

- It would be helpful to have Q1 to Q4 denoted again in at least one of the panels.

Done.

- In the caption you write "in-situ observation ... (non-binary)". I am confused ... so here you binned the ice products but not the evaluation data? Why? This is inconsistent.

Sorry that this was not clear. As described in section 4.1.4, it is imperative to perform a consistent comparison among the datasets. Since MASIE has an inherent cutoff threshold of 40%, to compare it with grASIP, AMSR2, and the in situ observations, these datasets must also be cutoff at 40%. Therefore, for all four datasets (grASIP, AMSR2, MASIE, and in situ observations) when SIC < 40% it is considered as having no ice, while when SIC >= 40% it is considered as having ice. To do this calculation, we need in situ observations that provide SIC, and thus we exclude the binary in situ observations from this comparison. This has been calrified in the text and figure caption.

L341/342: "but at this point framework" ---> Why? I doubt that this is a useful comparsison and that it provides a credible result.

Please see the responses to GC1 and GC3.

L348: "binned at ..." --> To me this looks as if this would result in 11 bins but Figure 6 contains 12 bins at both the x and the y-axis. Also the annotation with 10, 20, 30 ... does not fit well with the respective bin boundaries of 5, 15, 25, 35, 45% et cet. Please check and if need be correct.

Thank you for noting this. There was an issue with our inclusive lower and upper bounds. The figure, methodology, and text, have been updated. Now data are binned at 10% resolution throughout all concentration intervals.

L353-359: "Subsequently ... by AMSR2" --> I don't think that this step, particularly in this over-simplified fashion, adds value to what is shown in Fig. 6. I suggest you compute the overall difference and its standard deviation (or the RMSE) and to also compute the mean absolute difference. Both you can report in a separate table or in the text.

Thank you for this suggestion, we have updated the text to include this calculation. Note that the simplified version that we presented was the mean absolute difference, without normalization. The results of both the RMSE and the mean absolute difference (MAD) are presented in Table 4. Note that this calculation can be repeated: once for just the averages (i.e. the squares) and once for all the data points that go into Fig 6, which essentially represents the weighted RMSE and MAD. The final column represents the RMSE and MAD calculations just for the MIZ (20% – 80%).

L370+ / Figure 7: I don't find this additional parity plot useful. The information one can take from this figure one can as well simply take from Figure 6.

Respectfully, we disagree as to the use of this parity plot. This figure highlights the data available in the MIZ, and the resultant accuracy rate for grASIP and AMSR2 in this region. While this information is included in Fig. 6, it would be challenging to diagnose the exact accuracy in the MIZ from Fig. 6. Specifically, the 36/30 split for ice/no ice for grASIP and the 17/49 split for ice/no ice for AMSR2 is not quantifiable in Fig. 6; one can see that generally,

grASIP has more matchups above the one-to-one line and AMSR2 has more matchup below the one-to-one line in Fig. 6, but it is not possible to quantify how many correct matchups are present in each of these datasets from Fig. 6. Therefore, we have chosen to keep Fig. 7.

L404: "where the products most strongly disagree" --> Which you could again nicely derive from Fig. 6 by computing the mean SIC difference and the mean absolute SIC difference using the in-situ SIC range of 15-80%.

Thank you for this suggestion. This is now included in Table 4 and referenced in the text.

L406: I don't understand what this "accuracy rate" is. Are you computing the SIC difference? Possibly not because you seem to refer to the ice edge only. So what are you looking at here? The accuracy of which geophysical parameter? And why "rate"

No, we are computing how many match-ups in each bin are correct, and how many are incorrect. In other words, for each range of distances from the ice edge (e.g. 25 km – 50 km from the ice edge, in the ice), we ask how many times the product and in-situ asset agree that ice is present, and how many times they do not. This then provides an accuracy for that given distance bin. The calculation is repeated for each distance bin (e.g. 50 km – 75 km, 75 km – 100 km, etc.) and for each product (grASIP, AMSR2, and MASIE). This results in a binned accuracy as a function of distance from the ice edge. Please note that given the confusion around this figure, we have removed it from the text, as the results from the figure are shown similar in Tables 3 and 4.

L410-413: Sorry, but I don't understand what you did here. I see an accuracy rate given in percent at the y-axis (in %) but I don't know of which parameter and I see a distance from the ice edge in km (possibly the center of the grid cells are taken - even though I recall that you were reprojecting data onto a 0.05 degree grid ...).  But what do the curves tell me?

We apologize for the confusion; given that this figure has confused multiple readers, and given that the results it presents are similar to those provided by the parity analysis, we have decided to simplify the text by removing the figure and the discussion.

For the reviewer's reference, we did the following (now excluded from the manuscript).

1. We want to know how accurate each product is at recognizing the presence or absence of ice as a function of distance to the ice edge.
2. To do this we compute the distance of an in-situ asset to the ice edge in ASIP, in AMSR2, and in MASIE. Therefore, now each in-situ asset has three distances: a distance to the ASIP ice edge, a distance to the AMSR2 ice edge, and a distance to the MASIE ice edge.
3. Then, for each product, we perform the following analysis.
   a. For each 25 km bin defined by distance to the ice edge (e.g. for the 25 km – 50 km range into the ice, or the 50 km – 75 km range into open water), we find all available in-situ assets that fall in that distance range.
   b. Of those in-situ assets, we compute how often the product is correct about the presence or absence of ice for that in-situ asset.

i. So for example, if we had 10 in-situ assets that were between 25 km and 50 km of the ASIP ice edge in the ice, and ASIP reported 8 of these grid cells as having ice, then the 25 km – 50 km bin would have an accuracy rate of 80%.

4. Once this calculation is done for each product, and for each distance bin, we can plot the curves as shown in the original Fig. 9, which allows us to asses the accuracy of a product at determining the presence/absence of ice as a function of distance to that product's ice edge.

L422/423: I don't understand the purpose of this 3x3 pixel window smoothing. Why do you want to remove small-scale features? What is the motivation / scientific rationale behind this step?

We use 3 x 3 pixel window smoothing to remove small-scale features in order to do a large-scale/basin-wide intercomparison between ice edges. We now explain this in the text.

L423-426: Please check the scientific literature with respect to the ice edge delination as carried out by you. There should be several papers published that have done this (e.g. Cortenay Strong et al. "On the definition of marginal ice zone width", Journal of Atmospheric and Oceanic Technology, 34, 2017). You might want to check whether your idea is similar to their's and cite and/or check the existing literature for more examples to back up your approach better.

We have now added a paragraph describing a variety of methodologies used to define the MIZ (e.g. Strong et al., 2017; Strong and Rigor, 2013; Strong, 2012; Stroeve et al., 2016). We originally implemented a similar technique to the radial technique used by Stroeve et al. (2016), but thanks to the reviewer's comment we have simplified our approach to rely solely on the Stroeve et al. (2016) technique, as it is a more intuitive method than our original algorithm.

L450/451++: "This is likely ..." --> maybe yes, but not necessarily because at the ice concentration ranges (around 15% and around 40%) you are considering here, the melt pond fraction on the sea ice should be rather small because ice floes have disintegrated and quite some amount of the ice encountered might be brash ice.

Thank you for this comment. We note that here we were referring to snow and melt on the surface of ice as well as melt ponds. We have tidied-up the language to state that "melt ponds and snow and melt on the surface of the ice" instead of "melt water" could be causing a challenge for passive microwave measurements at low concentrations in summer.

I invide the authors to check the available literature about other possibilities to explain the observed discrepancies. There has been a study about why MASIE shows ice while other products don't, for instance.

Meier et al. (2015) compare passive microwave products with MASIE, and they demonstrate a similar result (but for sea ice extent (area) instead of distance). They attribute the discrepancy between MASIE and passive microwave, especially in summer months, as potentially due to 1) melt water on the surface of ice, which would result in an under-estimation of ice by passive microwave, 2) the presence of new, thin ice, which would result in an under-estimation of ice by

passive microwave, 3) the presence of new, thin ice that could be hard to distinguish for MASIE analysts, leading to an over-estimation of ice by MASIE, 4) the lack of clear imagery, that could make an analyst reluctant to shift the ice edge until a new image is available, which would result in an over-estimation of ice by MASIE, and 5) the higher-resolution nature of MASIE, which would result in an over-estimation of ice by passive microwave (since it would struggle to see openings in the ice near the coast). A discussion of this has now been included in the text.

In general, what should follow here is a discussion into the direction of the credibility of the approaches compared. Influencing factors are the grid resolution and/or the resolution of the input data. This applies to ASIP, MASIE and AMSR2-ASI. Please carefully check how ASI treats potential spurious ice along the ice edge due to the elevated weather effect one has to deal with at 89 GHz. If I am not mistaken, then the ASI algorithm is actually kind of a hybrid product where "bad" sea ice is filtered away by using other, coarser resolution SIC data.

A detailed discussion of potential sources of error and data limitations has been added to the text. Furthermore, a comparison of footprint sizes among data sets is presented in the new Methods section (3.4 Footprint size).

Thank you for your comment. We have added a brief discussion on algorithms to section 2.3.1. However, a detailed description of the ASI algorithm is outside the scope of this paper and we reference the reader to Melsheimer, 2024.

Melsheimer, C.: ASI Version 5 Sea Ice Concentration User Guide, https://data.seaice.uni-bremen.de/amsr2/ASIuserguide.pdf, 2024.

To respond to the reviewer's question: that is correct, ASI uses the lower frequency channels (with lower resolution; 18, 23, 37 GHz) to mitigate the increased weather effects due to the higher frequency 89 GHz channel through a series of filters that use gradient ratios between channels. Additionally, ASI uses the Bootstrap algorithm to set ASI SIC to 0% when Bootstrap SIC is less than 5%, as Bootstrap does not have as many problems as ASI with atmospheric processes, as it uses lower-resolution frequency channels (18 and 37 GHz).

Another issue you might want to discuss is the tendency for ice analysts to, as a first guess, take the conditions of the previous day - especially if there are not enough (high-resolution) satellite data of the day in question at hand. How often is the information given in the ASIP or MASIE product actually based on coarse resolution satellite data from passive microwave sensors (e.g. 25 km)?

We have added a discussion of this phenomenon to the text. Although we cannot quantify how often the information in ASIP and MASIE comes from passive microwave (e.g. 25 km resolution), we note that Meier et al. (2015) document an example where the MASIE ice edge did not change despite passive microwave changing. Similarly, Steele and Ermold (2015) show that the MASIE ice edge is more prone to loitering, or remaining in the same geographical location for multiple days in a row, when compared to a passive microwave ice edge. Both these examples would suggest that it is not often that analysts use only passive microwave to draw the ice edge. This has been added to the text.

Another issue not touched by you is the fact that ASIP uses polygons and that you are dealing with a sea ice concentration range. Neither the location and extent of the polygon nor the sea ice concentration range in these are overly well defined or FAIR in the sense that repeated analysis would result in exactly the same result; it is not transparent.

We agree that the manual analysis and range of SIC values in each polygon introduce sources of errors, as discussed in the new section 3.2 (Motivation for and conversion to binary ice/no ice). Specifically, given that polygon shapes are chosen by the analyst, and that polygons only exhibit concentration ranges, and not specific SIC values, this motivated our choice to retain a binary ice/no ice framework, instead of computing accuracy estaimtes as a function of SIC (see response to GC1 and GC3).

Finally, how much are ASIP maps generated in the sense to provide maximum safety for navigation and therefore - similarly to the various ice charts available - come up with a rather conservative estimate, likely tending to overestimate the true ice conditions for the sake of maritime safety?

ASIP ice analysts do not have this directive. Of course, implicit bias could result in a tendency to overestimate the true ice conditions if an analyst errs on the side of caution, but this is not a stated edict at ASIP. This has been described in the text.

L485-486: "Since the ... " --> Ok, but how much "hand-waving" is involved into drawing the polygons' boundaries in comparison to a well-defined 3.125 km gridded SIC product as provided from AMSR2 using ASI?

This is a good point and the statement has been removed.

L488/489: "...where they have been observed" --> exactly. So what is with, e.g., the next day, when there is no high-resolution information available but only a AMSR2 6 GHz 50 km footprint-based SST estimate because there are clouds? Such a day-to-day hetereogeneity is not helpful and combining different spatial scales of information requires particular care when it comes to assess uncertainties. I am pretty sure that the ASIP and to some degree also the MASIE product stitch different scale-observations together and the credibility of the data product can change quickly from one pixel to the next and from one day to the next.

Thank you for your comment; as this is outside the scope of the paper, the sentence has been removed.

L491: See my earlier comment about the work Worby and Comiso did. It is the Antarctic and it is ASPeCt. You must not use it to refer to ASSIST.

Thank you for catching this. We have updated the text accordingly.

L491/492: "recall that ... at that time" --> While this is true, the ship is moving during the 10-minutes observation time, hence elongating the observed area towards an elliptically shaped

region centered along the ship's track. In addition, if I am not mistaken, you did not compare single ASSIST observations but looked into daily averages?!?

That is correct, and we have added a sentence to the text to state that the ship could be moving during the 10-minute sampling window, thus increasing the area covered slightly. We looked into daily averages, as is detailed later in the paragraph, but it is worth noting the limitations on individual measurements before considering the limitations of the daily averages. Systemic biases in the individual observations will then feed back onto the daily averages.

L492-502: All true and possibly also discussed to some extent in Kern et al. (2019), right?

Thank you for noting this, some (but not all) of these points are discussed in Kern et al. (2019) and Bietsch et al. (2015). The citations have been added to the text.

L503-511: What I would strongly recommend is to suggest further evaluation of ASIP with independent observations of the sea ice conditions from Sentinel satellites (Sentinel-1 SAR and Sentinel-2 MSI). These provide a spatial representation of the conditions at the ice edge / in the MIZ and potentially would be a more solid basis for any further evaluation.

The suggested analysis is an interesting extension of the work, and would be an interesting study, but is outside the scope of this paper. We have now included a sentence in the Discussion section indicating that additional validation using SAR might be useful.

Editoral Comments / Typos:

L60: "Steffan" needs to be "Steffen"

Fixed.

L64: "imagery" --> "imagers"

Fixed.

Figure 1: I suggest to increase the size of the panels a bit to enhance readability. Alternatively, increasing the font size would help as well.

Done. We moved panel (e) below panels a-d, and we moved the details on the AMSR2 ice concentration binning into the body of the text, per reviewer 2's request.

L130: I guess this was August 21 and not August 12? Where was the image taken? Could you indicate that in one of the maps?

No, the image was taken on August 12. The Wave Gliders themselves did not have image capability, so the image was taken from the RV Ukpik during deployment. We chose to show ice maps on August 21 in order to show the tracks one week before and one week after an ice map,

to demonstrate the persistence of the ice tongue discussed in the text. This has been clarified in the text.

L221: As far as I know the two Kern et al. papers are dealing with both the Antarctic and the Arctic - especially the one from 2020.

Thank you, the text has been updated to state this explicitly.

Table 3: The way to specify inclusivity in values ranges would be [15% to 80%] or [0-40%[ or ]80 to 100%]

Thank you, the table has been updated accordingly (using the bracket vs. parenthetical notation for closed vs. open intervals).

L345: "double triple" --> typo

Fixed.

Figure 6:

Fonts at the legend should be larger.

There is no legend, but font size was increased for the full figure.

I suggest to change "% of time" to "count"

Thank you for the suggestion. We retain % of time, as the term "count" does not accurately represent the shading.

I also suggest to write "sea ice concentration" instead of just "ice" when denoting the axes.

Done.

Figure 8:

Please increase the font sizes.

Done.

Figure 10: Please provide the unit of the distances.

Done.

---

## Author Comment (AC2)

General comments

GC1: The authors compare how well the NOAA Alaska Sea Ice Program (ASIP) daily ice charts, along with another operational ice extent map and a passive microwave sea ice concentration product, match up with sea ice concentration estimates from visual shipboard observations. I have some concern that a casual reader of the paper will see a statement like "ASIP's overall accuracy rate of 95.7%..." and use it without reference to the limitations of the validation method that the authors are aware of. The authors will improve the manuscript by tightening and clarifying the presentation, especially when describing the ASIP and MASIE products and how they are "parsed".

Thank you for your comment and for these helpful suggestions to tighten up the language in the paper. We have changed the terminology and explained in detail how the data are read and converted from SIGRID polygon information to gridded SIC maps (the process we previously referred to as "parsing and gridding). Furthermore, we have updated the manuscript to be more explicit about the meaning of operational ice products.

To be mindful of the reviewer's concern that someone might see the accuracy rates/percentages and not consider the limitations of the in situ data used for validation, we have updated the manuscript to present accuracy rates with the caveat: "in situ observations, which do not cover all grid cells and all times, and are thus not comprehensive", or some variation of that phrasing.

GC2: Sea ice charts are often the best information available to researchers as well as to those operating in polar waters. Yet, charts are underused by the research community, because researchers are often unfamiliar with them and have no way to evaluate their quality. Research papers that attempt to quantify the accuracy of operational products are few. That makes this one especially valuable, if the presentation is improved.

We thank the reviewer for their kind words and for the enthusiasm for our study. We are grateful for their helpful comments, which have strengthened the manuscript.

Specific comments

SC1: In the abstract, the authors write "…we show that the similarity in performance among products is due to the in-situ asset distribution, as most in-situ observations are far from the ice edge in 20 locations where all products agree." This statement would seem to discount their results. It illustrates why I think the manuscript needs at least a paragraph in section 2.1 describing how analysts make the charts, and a section with at least a few sentences describing how "ice edge" is defined and drawn, if it its drawn, in or using the ASIP, AMSR2, and MASIE products for the purposes of this study.

We have updated the abstract to read: "we show that the similarity in performance among products is partly due to the deficiencies in the in situ asset geographical distribution, as most in situ observations are far from the ice edge in locations where all products agree".

We have included a description of how analysts draw polygons in section 2.1. A section describing how the ice edge is defined in each product has been included (3.3, in the new Methods subsection recommended by reviewer 1).

SC2: This discussion of how "ice edge" is defined, drawn, and used in the three products should come ahead of the Results section. Section 5 Discussion has some of this, but having the information in a stand-alone section and moving it forward will help readers understand how the differences in where products put an ice edge may arise. As it is, the authors begin using "ice edge" without an explanation. I think of the ice edge as a contour line. Do the authors create a contour line in 0.05° gridded versions of AMSR2 SIC fields, MASIE 1 km binary ice/not ice fields, and ASIP polygons containing ice concentration ranges?

Thank you, we have added a section to the Methods section that describes how the ice edge is defined in each product (section 3.3).

SC3: Each product sets a different-sized area within which SIC is estimated. The AMSR2 grid cell size may be 3.125 km but the SIC algorithm integrates brightness temperature information from frequency channels that have different footprint sizes and shapes. The ASIP analyst looks at satellite imagery and draws a polygon around ice that looks homogenous or ice floes that are fairly evenly distributed, and labels it with an ice concentration range. Each polygon is different. The USNIC analyst that draws the IMS product used by MASIE estimates which 1-km grid cells cover areas with more than about 40% ice and labels them "ice", using a variety of satellite and other data sources. Finally, the ASSIST observation is for an area within 1 nm of a ship, although visibility may limit this, as the authors note. Describing all this in one place will help the reader have a fuller picture of how differences in ice edge position arise.

Thank you for raising this point. To address the issue of differences in footprint sizes and temporal resolution (also raised by reviewer 1), we have created a section (3.4 Footprint size) where these nuances are presented upfront and together, before the analysis.

SC4: I don't think it would be particularly useful even if it were possible to come up with a rigorous accuracy estimate for these products. I think it's more important to understand how they are made and the strengths and limitations of each. The authors note that the ASIP product puts the ice edge further south than MASIE or AMSR2, and is more accurate when it does so, judging by shipboard obs. If you are a researcher that needs to know how likely it is that ice at any concentration will be present at some location off the coast of Alaska, then the ASIP product is your best choice. A tighter, more carefully written Discussion section up front will help more researchers understand that choice.

We have updated the manuscript to clarify that the accuracy rates are only a function of the in situ observations and are not comprehensive (see response to GC1). Further, we have added a new Methods section (per reviewer 1's request) and included two discussions of how ASIP, AMSR2, and MASIE are different: section 3.3 compares and contrasts how the ice edge is defined in each product, section 3.4 compares and contrasts the footprint size of each product.

The Discussion section has been updated to say that ASIP's ice edge is generally further south, and is generally more accurate as judged by in situ observations. Thus, we recommend the data for scientific and operational stakeholders alike.

SC5: The Discussion section also needs something on why MASIE and the hi-res AMSR2 from Bremen were chosen. Note that MASIE is not itself an operational product but is a reformatting of the USNIC IMS operational product. I assume MASIE was chosen because it is easier to work with than IMS and offers a unique daily high-resolution map of ice extent.

We have added a discussion of why we selected these two products. We chose to put this earlier in the manuscript, in the Data section, so that readers understand upfront why these products were chosen. Now the text reads: "MASIE was chosen, instead of the USNIC IMS operational product, for example, because it offers a unique daily high-resolution map of ice extent, is provided in an easy-to-use gridded format, and represents a product commonly used in the scientific literature that is generated following similar methodology to the grASIP dataset."

SC6: The USNIC MIZ product (U.S. National Ice Center, 2020) is another daily product that shows a 10% and 80% SIC contour. The authors could consider working with it as an alternative or in addition to MASIE.
U.S. National Ice Center (2020). U.S. National Ice Center Daily Marginal Ice Zone Products, Version 1 [Data Set]. Boulder, Colorado USA. National Snow and Ice Data Center. https://doi.org/10.7265/ggcq-1m67.

Thank you for suggesting this analysis. While it would be interesting to include the USNIC MIZ product in this study, our goal was not to provide a comprehensive comparison between all or many SIC products and instead present a comparison between a few representative datasets. Thus it is outside the scope of this manuscript. This is because we aim to show the reader three types of data and (a) how they compare and contrast and (b) how someone could repeat this analysis for a different selection of datasets.

SC7: It would be helpful to mention that USNIC charts also cover the region covered by the ASIP charts, and have some words about how they compare, as RC1 noted.

Done.

SC8: After years of working with ice chart products along with satellite data, I strongly agree with the authors concluding statements about the value of ASIP products for scientific studies.

Thank you for your enthusiasm!

More specific comments follow.

L9: "….we present a new SIC product…" Please clarify exactly what the new product is and how it differs from the ice charts that are available on https://www.weather.gov/afc/ice. The text isn't clear on this.

We have now updated the abstract to specify that "we present a newly-gridded SIC product generated from data from the…". We have also clarified that these data are different (source vs. gridded fields) in Section 2.1.

This raises an interesting point that we had not previously considered. We fundamentally modify the ASIP source data when we grid it and convert the concentration ranges into SIC values. For this reason, it seems appropriate to use a different name to distinguish the data from their source data stream, as MASIE did. For this reason, we have updated the text to refer to grASIP (Gridded ASIP), instead of ASIP.

L11: Does EGU prefer "in-situ" to "in situ"?

Thank you for pointing this out. According to the submission policies, they ask that Latin phrases not be hyphenated. We have updated the manuscript accordingly.

L16:   Consider rewriting as " … and (iii) a product available from the National Snow and Ice Data Center (MASIE) that originates with the US National Ice Center (USNIC) operational IMS product."

NSIDC archives both products, and both should be cited correctly, and listed in the References section.  Here are the MASIE and IMS citations in APA style:

U.S. National Ice Center, Fetterer, F., Savoie, M., Helfrich, S. & Clemente-Colón, P. (2010). Multisensor Analyzed Sea Ice Extent - Northern Hemisphere (MASIE-NH), Version 1 [Data Set]. Boulder, Colorado USA. National Snow and Ice Data Center. https://doi.org/10.7265/N5GT5K3K.

U.S. National Ice Center (2008). IMS Daily Northern Hemisphere Snow and Ice Analysis at 1 km, 4 km, and 24 km Resolutions, Version 1 [Data Set]. Boulder, Colorado USA. National Snow and Ice Data Center. https://doi.org/10.7265/N52R3PMC.

Done.

It's important that readers understand that NSIDC is not an operational center, and MASIE is not an operational product, in contrast to USNIC and IMS.  It would be helpful to say what is meant by the term "operational" as used in this paper.

The fact that MASIE is not an operational product is now stated explicitly in Section 2.3.2. We now provide an explicit definition for operational ice products in the Introduction.

L31: "MASIE has by definition no information at SIC < 40%."  That's not entirely true. One could regrid MASIE to some larger grid-cell size, and end up with larger grid cells with less than 40% SIC.

Thank you for this point. While we agree that one can always blur out an ice edge by moving to larger and larger grid cells, the product itself does not provide information beyond a 40% cutoff.

That said, this is important when we re-grid MASIE to grASIP grid for the analysis in section 4.3 and introduce non-binary values along the ice edge. We now explain that the SIC = 0.5 contour from this gridding is used, to be as true to the source data as possible.

L35-131 The Introduction section could be shortened and tightened up a lot. Omit needless words.

We have tightened up the wording in the introduction section. However, given the suggestions for this section from both reviewers, we have slightly increased the length of the introduction, as opposed to shortened it.

L58: Lohanick is misspelled.

Thank you for catching this, it is fixed.

L59: It would help users understand better if written "This leads to an underestimation of sea ice concentration, which in turn results in an underestimation of sea ice extent…"

Done.

L63-67: While the first method describes using a processing algorithm on satellite data, the second method describes how a human might draw a chart. Different word choices might better get across the manual nature of drawing operational charts, e.g "an analyst manually synthesizes the information in satellite imagery …" ; "Operational maps as drawn…"

We have modified the word choice to emphasize the manual nature of this second class of ice products. We also take this opportunity to define what we mean by operational ice products.

L77: Consider citing the CIS data so that others can easily find it:

Done.

Canadian Ice Service (2009). Canadian Ice Service Arctic Regional Sea Ice Charts in SIGRID-3 Format, Version 1 [Data Set]. Boulder, Colorado USA. National Snow and Ice Data Center. https://doi.org/10.7265/N51V5BW9.

Done.

L80-81: here, please make the clarifications and add the citations that I noted with respect to the abstract. Also, I hear that USNIC prefers USNIC to NIC these days.

Done. We also have updated the manuscript to use USNIC, instead of NIC.

L87-88: This is the first mention of color code, egg code, and WMO standards. A few things are incorrect. "Egg code" is not a WMO standard, rather, it is a shorthand way, taken from the egg shape of the labeling symbol, that analysts use to refer to how a polygon in an ice chart is

labeled. The ice information inside the egg symbol would be in SIGRID, which is a WMO format. While SIGRID is used by ASIP, USNIC, and other ice services to describe the ice within each polygon, egg codes are not used much anymore. (Danish Meteorological Institute charts are an exception. See https://www.bsis-ice.de/IcePortal/)

I would avoid using "egg code" entirely.   Instead, just briefly mention that you are following ASIP and international ice chart convention in using the WMO color code and descriptors for characterizing ice concentration ranges in your presentation of AMSR2 and MASIE sea ice concentration as well as for ASIP.    You can cite WMO Sea-Ice Nomenclature (WMO, 2014):

World Meteorological Organization (WMO). 2014. WMO Sea-Ice Nomenclature. Volume 1 - Terminology and Codes, Volume II - Illustrated Glossary, Volume III - International System of Sea-Ice Symbols. Fifth Session of Joint Commission on Marine Meteorology (JCOMM) Expert Team on Sea Ice. WMO Publication No. 259.

and perhaps Manual of Standard Procedures for Observing and Reporting Ice Conditions (MANICE) (Env. Canada, 2005)

Environment Canada. 2005. Manual of Standard Procedures for Observing and Reporting Ice Conditions (MANICE). Issuing authority: Assistant Deputy Minister, Meteorological Service of Canada.

Thank you for the clarification on the origins of the term "egg code". We have updated the text to avoid using the terminology "egg code" and instead do as suggested by the reviewer, by explaining that we are using WMO convention for color codes and ice descriptors. The reader is then referenced to both the citations listed here, as well as section 2.1, which has been updated following this comment.

L89: Include acronym here (SASSIE) if correct to do so.

Done.

L120: Remove "operational" here.

Done.

L123-131 and Figure 1:  The images here need to be MUCH bigger. Delete the photo (e) if necessary in order to enlarge the rest of the figure.  Also please clean up the text about WMO and eggs, and provide the information on how AMSR2 data are binned somewhere in the main text, not the caption.

We have shifted panel (e) below the SIC maps (a-d), in order to enlarge panels a-d, and we have removed the mention of egg code. The AMSR2 SIC binning information has been moved to the text.

L143-144: Suggest you include just one WMO reference, this one, at the end of the second sentence:

World Meteorological Organization (WMO). 2010. SIGRID-3 : A Vector Archive Format for Sea Ice Charts. Intergovernmental Oceanographic Commission. First edition: 2004. JCOMM Technical Report No. 23, WMO/TD-No. 1214: Https://library.wmo.int/index.php

Done.

L148: Suggest changing "but is analyzed from imagery over the preceding 24 hours." To "based on imagery acquired over the preceding 24 hours."

Done.

L154-180:  Please consider my general comments when editing this section, and describe how a person draws polygons. The word "implement" is misleading. Also, it's not always clear what is meant when the word "parse" is used.  Consider choosing other words to describe the process of gridding the ASIP polygon SIC information onto a grid in some projection.

I suggest you reference the following data set somewhere in Section 2.1.  As with the ASIP product, when we made it, we needed to convert SIC information in shapefile polygons to gridded fields of SIC.  The User Guide for the product describes the process we used.  There are so few products of this kind that it would be helpful for readers to know about this one as well:

U.S. National Ice Center. 2020. U.S. National Ice Center Arctic and Antarctic Sea Ice Concentration and Climatologies in Gridded Format, Version 1.  Boulder, Colorado USA. NSIDC: National Snow and Ice Data Center. https://doi.org/10.7265/46cc-3952.

Thank you for the helpful comments to tighten up word choice. We have added an explanation for how an analyst draws a polygon. The word implemented has been replaced with drawn. Instead of using the word parse, we now explain that the data are read, projected, and the alphanumeric string corresponding to ice concentration and form are converted to numerical information. Then the data are gridded and SIC is computed. We also now refer to the above citation to demonstrate that this has been done similarly for other datasets.

Figure 2 (c) and (f): Suggest using a step color bar.

Done.

L255: Remove "operational".

Done.

L260: Clarify what "it" refers to.

Done. Please note, we have expanded on this section at reviewer 1's request.

L267-276: Please edit, taking my general comments and other specific suggestions into account.

Done.

L278: "We compare satellite SIC to in-situ observations…." What does "satellite SIC" refer to? Is it just the AMSR2 25km and 3.125 km products? Please specify. MASIE and the ASIP products should not be referred to simply as "satellite SIC". Perhaps "gridded SIC fields" is a better choice?

This has been clarified.

L279: Further confusion about satellite products here. Line 280 refers to the "nearest satellite pass", but earlier, AMSR data were described as a daily field and not as swath data. Is the time of the nearest satellite pass known? "Satellite grid cell": does this mean the AMSR2 SIC grid cell?

Thank you for identifying this confusion. We have clarified the text to refer to SIC products/maps. While we do not know the specific time of the nearest pass, we take the timestamp from the daily gridded field in the SIC product. The reason this still matters is because there are cases when an in situ observation is > 12 hours from a SIC map (this is especially true for ASIP matchups, as ASIP does not have daily maps for the entire record).

L285: Here, "three satellite products" implies that "satellite products" is being used to refer to the MASIE and ASIP fields and not just the AMSR2 data. Please choose words other than "satellite products" and clarify the reference to "nearest satellite pass".

Done. We now refer to the datasets as "SIC products/map".

L292: Section 3.1 is titled "Satellite products compared with in-situ observations". Change "satellite products".

Done. This now reads "Ice maps compared with in situ observations".

L295: Suggest addition to read "…defining the ice edge in passive microwave products…"

Done. Note, this sentence has been moved to earlier in the manuscript to the methods section (3.3 Defining the ice edge).

L340 and section 3.1.2: It's not surprising that AMSR2 under-predicts the presence of sea ice, and it's helpful to see it demonstrated by the comparison with shipboard obs and with the ASIP operational charts here.

Agreed, thank you.

Figure 7: I agree with RC1 that these figures can be much smaller and font bigger.

Done.

Figure 8: Suggest including only one (bigger) legend and making the MASIE edge a different color.

Done. Thank you for pointing out the red/green issue. We have updated the AMSR2 to a dark red and MASIE to a yellow/orange. The figure has been updated to have one large legend. Note, to maintain consistency throughout the manuscript, we have also updated Fig. 6 and Table 3 to use this color scheme.

L434: Does the word "difference" belong after "distance"?

We have clarified the meaning by re-wording the text to read "ice edge distance between products".

Figure 10: the Y axis needs units.

Done.

L468: The phrase "lowest common denominator" doesn't work well here.

We have removed this section and instead describe how the comparisons are made in the new methods section (3.1 Parity analysis, paragraph 3).

L479-486: This is important information that should come earlier.

This has been shortened and moved earlier in the manuscript, to section 4.3.

L 485: Suggest addition to read "…ASIP product is a vector shapefile that is not provided…"

Per reviewer 1's comments, we have removed this statement.

L494:  "grid cell" would be better than "pixel"

Done.

L496: Suggest instead of "the broader pixel" using "are reflected in the SIC grid cell"

Done.

L520:  IMPORTANT- Should read "…grateful to all the analysts at ASIP and at USNIC".

Thank you, done.

L558: This should read:

U.S. National Ice Center, Fetterer, F., Savoie, M., Helfrich, S. & Clemente-Colón, P. (2010). Multisensor Analyzed Sea Ice Extent - Northern Hemisphere (MASIE-NH), Version 1 [Data

Set]. Boulder, Colorado USA. National Snow and Ice Data Center.
https://doi.org/10.7265/N5GT5K3K.

Done.

L585: Please include DOI or link for this report.

Done.

L598: Please include DOI or link for this report.

Done.

L602: Please include DOI or link for this report.

Done.

---

## Author Response (AR2)

Review of

National Weather Service Alaska Sea Ice Program: Gridded ice concentration maps for the Alaskan Arctic - Revision 01

by

Pacini, A., et al.

Summary: I don't give another summary, see my review of the first version of this manuscript

General Comments:
I don't have any general comments and concerns anymore. The authors did a great deal in improving the manuscript. I only have a few specific comments where I would like the authors to invest some thought and accordingly, if they see fit, change their wording to explain a few things with even more clarity. I also found a few typos.

Thank you for your comments and suggestions, which have strengthened the manuscript. Please find our detailed responses in blue below.

Specific Comments:
L304-306: Please provide a reference to where this step is described in the context of the AMSR2 3.125 km sea ice concentration product you used.

Done. We have added a reference to Melsheimer, 2024.

Melsheimer, C.: ASI Version 5 Sea Ice Concentration User Guide, https://data.seaice.uni-bremen.de/amsr2/ASIuserguide.pdf, 2024.

L313-315: "Many studies ... 2015)." --> I agree, several studies exist that looked at the ASI algorithm sea ice concentration data at 6.25 km or coarser grid resolutions. But no study does exist yet which - similarly to Kern et al., 2019 / 2020 / 2022 provided a systematic evaluation of the AMSR2-based ASI sea ice concentration data at 3.125 km. The paper of Beitsch et al. (2014) which you cite in the next sentence demonstrates that this fine resolution product is capable to resolve small-scale sea ice concentration variations better than coarser resolution products. But this is an example for an application rather than an independent evaluation. To my knowledge, such a work is still pending and has not yet been performed. Hence, in contrast to basically all other existing products one can obtain, the 3.125 km product is super nice, but is not yet evaluated.

Thank you for your comment; we agree that the Beitsch et al. (2014) demonstrates that the 3.125 km product is useful for a specific application, but no study exists, to our knowledge, that performs an independent evaluation of this product. It would be great to see this done in the future.

L348-350: Could you remind the reader why you chose 20% for grASIP and AMSR2 SIC data

while the MASIE product uses 40%. Possibly you mentioned already earlier in the manuscript that it does not make a difference whether one uses 20% or 40%? Particularly, in light of your note in line 289 where you said that saildrone SIC observations rarely report values > 40% because the saildrones are not designed to operate in ice-covered waters would - in my eyes - rather call to use 40% as a threshold. With that the saildrone observations would mostly be an indicator of open water conditions, yes, but this seems more reasonable to me than assigning SIC between 20% and mostly less than 40% SIC as 100% ice.

Reading the paper and coming to the results sections reveals that you actually did use other thresholds than 20% quite often. I am wondering whether you could change the information given in these lines, in such a way that the 20% threshold appears to be less strictly used. As you can see from my comments / questions I got puzzled a bit.

We have updated the text to state that "This conversion is done for a variety of SIC values. Unless otherwise stated…"

L396: "the 0.5 contour is chosen" --> I am not entirely sure what you mean by this. Do you mean that you used the half-way-distance between the centers of two neighboring 0.05 degree grid cells with one showing ice (aka a value of 1) and the other one showing no ice (aka a value of 0) to delineate the ice edge position in the vicinity of these two grid cells?

Correct.

L617-620: What you state in this paragraph is backed up by the results of your study, but I was wondering whether you could perhaps mention the time period for which your results are obtained - basically mid May to October - aka spring/summer/fall.

Done.

L647/648: "imagery may ... the ice pack" --> I find this combination of statements a bit contradictory. First you state, that there might be no images available for a day or even a series of days which means that the ASIP sea ice information is not updated. But then you state that ASIP is better suited to understanding the daily state.

The argument that we are making is that on a day with available imagery, then ASIP represents high resolution SIC information and provides a high-resolution daily estimate of the ice pack. Of course, if imagery is not available, then that specific day is based off of fewer input data sources.

L660-662: "Furthermore ... details.)" --> This is not entirely true because for the ice pack and also for "fair weather conditions" the AMSR2 SIC is primarily based on the high-resolution channel with near 3 km footprint size. The coarser resolution channels only come into play for the open water and/or cases where severe weather causes the original ASI-algorithm SIC to fail and to produce too high SIC values which are then filtered out using the SIC information derived from the coarser resolution satellite sensor channels.

We agree, but we maintain that these coarser resolution channels are used in certain scenarios, as the reviewer notes, and therefore as a whole, AMSR2 SIC is a blend of various channels.

Editoral Comments / Typos:

Table 2: The saildrones entry of 2019 has a typo: "Octo." --> "Oct."

Done.

L256: "Bietsch" --> "Beitsch"

Done.

L331: Helfirch --> Helfrich

Done.

L381: "are often order" ? Did you perhaps mean "are of the order of"

We have updated the phrasing to say "which are often on the order of 20%".

L448: "estaimtes" --> "estimates"

Done.

L448: "exclude the binary ..." --> which essentially means that you exclude the saildrone data, correct? You could write this accordingly.

Done.

L624: "is larger the" --> "is larger than the"

Done.

L630: "the Ice Watch" --> the ship-based"

Done.

L632: "Bietsch" --> "Beitsch"

Done.